# Trajectory-Based Neural Darwinism in Convolutional Neural Networks: Variation, Competition, and Selective Retention

## Abstract

Understanding how artificial neural networks develop and stabilize internal representations remains a central challenge in deep learning. Motivated by Edelman's theory of Neural Darwinism, we investigate whether competitive, selection-like dynamics emerge during training and how they shape robustness and specialization. We introduce a unified trajectory-based Darwinian framework—the Neuron Darwinian Dynamics System (NDDS)—which is inspired by Darwinian principles of survival and selection, enabling the analysis of neuron activations, weights, and representational paths across diverse architectures and datasets. We conduct two complementary analyses: ablation experiments demonstrate that networks maintain accuracy under extensive neuron removal, revealing strong redundancy, yet exhibit sharp performance collapse beyond a critical threshold, identifying task-critical subsets. Dynamic trajectory analyses further reveal consistent evolutionary patterns: neurons categorized as survived sustain coherent representational trajectories, stronger weight norms, and higher activations, whereas eliminated neurons stagnate toward representational silence. Overall, these results support a Darwinian perspective on representation learning: CNNs achieve robustness through redundancy at early stages and progressively consolidate specialized neurons that underwrite stable, task-relevant representations.

## 1 Introduction

The success of deep learning is often attributed to its ability to construct hierarchical feature representations Chizat & Netrapalli (2024); Banerjee (2025), yet the mechanisms that govern representational stability and neuron specialization remain only partially understood. Prior work has primarily emphasized optimization dynamics or information-theoretic principles Butakov (2024), while comparatively limited attention has been paid to competitive processes unfolding at the level of individual neurons. In neuroscience, Edelman's theory of Neural Darwinism proposes that neuronal populations evolve through variation, competition, and selective retention, thereby forming stable yet adaptable circuits. Building on this perspective, we investigate whether analogous competitive dynamics emerge in artificial neural networks and how they shape robustness and specialization.

Motivated by this, we introduce a unified trajectory-based Darwinian framework—the **Neuron Darwinian Dynamics System (NDDS)**—which formalizes neuron evolution in convolutional architectures through the lens of survival and selection. NDDS integrates trajectory-based analyses of representational dynamics, layer-wise inspection of activations, weights, and embeddings over training, together with controlled ablation to rigorously quantify representational resilience. This integrated view enables systematic comparison of neuron-level dynamics across models of varying depth and dataset complexity. Our experimental evaluation covers a spectrum of architectures and datasets, beginning with a three-layer MLP on MNIST and progressively extending to ResNet-18 on CIFAR-10, VGG-16 on CIFAR-100, and ResNet-50 on Tiny-ImageNet. Across these settings, neurons are categorized into survived, eliminated, and other groups according to long-term representational stability, providing a consistent lens for evaluating functional contributions. From an evolutionary perspective, the results reveal that different layers impose distinct selective pressures on neurons. Shallow layers exhibit highly variable and unstable trajectories, resembling an early exploration phase. Middle layers increasingly differentiate neurons into those maintaining sustained activity

and those drifting toward quiescence, suggestive of emergent selective filtering. Deep layers show a tendency toward contraction, where a relatively compact subset of neurons retains higher activation while others decline. These observations are consistent with Darwinian dynamics of variation and selection. Ablation studies further corroborate this interpretation, showing robustness under moderate perturbation and sharp collapse once the selectively retained subset is disrupted. We restrict our analysis to Convolutional Neural Networks in this work, as their hierarchical structure and well-studied representational dynamics provide a controlled and interpretable setting for isolating neuron-level evolutionary mechanisms. In contrast, Transformers introduce attention-mediated interactions and layer normalization effects that confound neuron-level attribution, making them less suitable for our initial theoretical analysis. Collectively, these findings suggest that CNNs achieve robustness and representational specialization not solely through gradient-based optimization, but also through emergent neuron-level competition that parallels Darwinian selection principles.

## 2 RELATED WORK

### 2.1 ON NEURAL NETWORKS ANALYSIS

A large body of work has investigated how neural networks form and consolidate internal structure, spanning pruning, representational similarity, loss geometry, and interpretability. Pruning studies demonstrate that overparameterized models contain trainable sparse subnetworks, with the Lottery Ticket Hypothesis Frankle & Carbin (2019) and its extensions Liu (2019); Sanh (2020); Lee (2019); Evci (2020); Morcos (2019) showing that subnetworks can be identified via sensitivity measures Lee (2019), dynamic rewiring Evci (2020), or transfer across tasks Morcos (2019). Representation analyses such as SVCCA Raghu (2017) and CKA Kornblith (2019) reveal convergent layerwise structures, while neural tangent kernel theory Jacot (2018) and deep linear dynamics Saxe (2014) provide analytic descriptions of training. Geometric studies show low-loss mode connectivity Garipov (2018); Draxler (2018) and neural collapse phenomena Han (2022), connecting optimization to generalization. Interpretability methods including Network Dissection Bau (2017), TCAV Kim (2018), Integrated Gradients Sundararajan (2017), and SHAP Lundberg & Lee (2017) further expose concept-level features, while symmetry and re-basin analyses Ainsworth (2023) link parameter permutations to solution geometry. Finally, work on large-batch training Keskar (2017) and dynamical isometry Pennington (2017) elucidates how optimization biases shape solution quality. Taken collectively, these perspectives highlight redundancy, convergence, and selection-like pressures in neural networks, aligning with our Darwinian view of neuron-level competition.

### 2.2 NEURON DARWINIAN

The conceptual foundation for Darwinian mechanisms in neural systems was laid by Edelman's theory of neuronal group selection, which frames brain function as variation among neuronal populations, selective reinforcement of circuits, and inheritance of stable connectivity patterns Edelman (1987). Inspired by this paradigm, recent advances in artificial networks embed analogous variation–selection processes across computational scales, challenging the dominance of gradient-only optimization. Du et al. reinterpret late-epoch backprop-trained models as "ancestral genomes" and evolve offspring via differential evolution to reduce overfitting and accelerate inference Du (2024). At the neuron level, NeuroFS dynamically prunes and regrows inputs under synaptic-plasticity constraints to maintain adaptability under sparsity Zahra (2023). In dynamical systems, Czégel et al. show Darwinian neurodynamics in reservoir computing, where activity patterns are imperfectly copied and fitter variants selected, yielding emergent combinatorial problem solving Czégel (2021). Evolutionary processes also benefit spiking models: Shen et al. evolve excitatory–inhibitory circuits via spike-timing–dependent plasticity, achieving strong CIFAR-10 and ImageNet performance Shen (2023). At the architectural scale, Shafiee et al. encode heritable "DNA" for evolving compact offspring networks Shafiee (2018), while Chen et al. propose OPNP, a gradient-sensitivity–based pruning scheme that improves out-of-distribution robustness by selecting fitter neurons and parameters Chen (2023). Collectively, these works demonstrate a convergent trend: embedding variation–selection mechanisms across synaptic, dynamical, and structural levels to improve adaptability, sparsity, and generalization beyond gradient descent. We extend this trajectory with a neuron-level temporal analysis framework that tracks activation trajectories to distinguish "survived" from "eliminated" neurons, providing direct empirical evidence for Neural Darwinism in modern deep learning.

## 2.3 NEURON TRAJECTORY

Recent work increasingly examines neuron trajectories—the evolution of individual activations or weights across layers and time—as a lens on training dynamics, interpretability, and generalization. Fu et al. formalize learning trajectories and derive generalization bounds tied to their complexity Fu (2023). Pesme and Flammarion analytically characterize gradient-flow paths in two-layer diagonal networks, showing convergence through successive saddles to minimal-norm solutions Pesme & Flammarion (2023), while Han et al. connect MSE training to the emergence of neural collapse by analyzing proximity and dynamics along the central path Han (2022), and Ahn links threshold-like neuron emergence to edge-of-stability dynamics Ahn (2023). In mechanistic interpretability, Conmy et al. introduce ACDC to extract activation subcircuits via trajectory-based graph discovery Conmy (2023), and Syed et al. apply attribution patching along activation paths to reveal causal transformer subcircuits Syed (2024). Beyond static analysis, Li et al. adapt trajectory forecasting (AMAG) to predict future neuron activity Li (2023), while spiking models leverage trajectory-inspired optimization to reduce firing load without loss of accuracy Shi (2024); Shen (2024). Together, these studies establish neuron trajectories as a unifying construct linking optimization dynamics, circuit discovery, and functional efficiency in modern networks.

## 3 METHOD

We formalize neuron evolution during training as a continuous-time dynamical system driven by both optimization gradients and intrinsic information-theoretic pressures. Intuitively, we treat each neuron as an evolving agent whose state is not only determined by its parameters but also by how it responds to data and gradients. This perspective allows us to study neural computation through the lens of dynamical systems and Darwinian selection Saxe (2014); Mei (2018); Chizat & Bach (2018).

Let a neural network $f_\theta : \mathcal{X} \to \mathcal{Y}$ consist of layers $\{L_k\}_{k=1}^D$, where layer $L_k$ contains neurons $\{a_i^{(k)}\}_{i=1}^{n_k}$. Each neuron is parameterized by a weight vector $w_i^{(k)} \in \mathbb{R}^{d_{k-1}}$, bias $b_i^{(k)} \in \mathbb{R}$, and activation function $\sigma$. Its activation at time $t$ is:

$$a_i^{(k)}(x,t) := \sigma\left(w_i^{(k)}(t)^\top h^{(k-1)}(x,t) + b_i^{(k)}(t)\right), \tag{1}$$

where $h^{(k-1)}$ is the output from $L_{k-1}$ and $h^{(0)} = x$. Thus, activations evolve jointly with weights and reflect both optimization and stochastic fluctuations Schoenholz (2017); Poole (2016).

### 3.1 NEURON DARWINIAN DYNAMICS SYSTEM (NDDS)

**Definition 3.1** (Neuron State Vector). To make this evolution explicit, we introduce the *neuron state vector*, which concatenates its trainable parameters, average activity, gradient statistics, and information-theoretic descriptors:

$$\psi_i^{(k)}(t) := \left[w_i^{(k)}(t),\ b_i^{(k)}(t),\ \mu_i^{(k)}(t),\ g_i^{(k)}(t),\ \mathcal{I}_i^{(k)}(t)\right]. \tag{2}$$

Here we explicitly define each component and its domain/estimation modality:

$$\mu_i^{(k)}(t) := \mathbb{E}_{x\sim\mathcal{D}}\left[a_i^{(k)}(x,t)\right], \tag{3}$$

$$g_i^{(k)}(t) := \mathbb{E}_{x\sim\mathcal{D}}\left[\frac{\partial\mathcal{L}(x)}{\partial a_i^{(k)}(x,t)}\right], \tag{4}$$

$$\mathcal{I}_i^{(k)}(t) := \text{(differential) entropy proxy of the marginal law of } a_i^{(k)}(\cdot,t). \tag{5}$$

We emphasize estimation modality: expectations are taken with respect to the data distribution $\mathcal{D}$; in practice they are approximated by empirical estimates over mini-batches. Throughout we reserve the symbol $\mathcal{L}(x)$ to denote the per-example loss.

The evolution of each neuron is then modeled as a differential equation:

$$\frac{d}{dt}\psi_i^{(k)}(t) = \mathbf{F}_\theta^{(k)}\left(\psi_i^{(k)}(t), \mathcal{D}, \mathcal{L}\right), \tag{6}$$

where $\mathbf{F}_\theta^{(k)}$ captures the joint effect of gradient-descent-like updates and intrinsic representational dynamics. This abstraction allows us to borrow tools from dynamical systems theory to analyze stability, convergence, and diversity of neurons Achille & Soatto (2018b).

**Assumption 3.2** (Smooth and Bounded Dynamics). We adopt a hypothesis compatible with practical discrete optimization. The parameter trajectory $\theta(t)$ is assumed to be absolutely continuous and piecewise $C^1$ in $t$ (so that it admits a time-continuous interpolation), and $\mathbf{F}_\theta^{(k)}$ is locally Lipschitz in $\psi$ on trajectories of interest. This formulation explicitly permits discretization effects arising from SGD and non-smooth activations (e.g. ReLU) by interpreting derivatives in the sense of absolutely continuous interpolation or Clarke subgradients when necessary. We assume standard smoothness and boundedness conditions on interpolated trajectories; detailed assumptions and discretization–continuum error bounds are deferred to Appendix

**Assumption 3.3** (Local Gaussianity of pre-activations and diagnostic protocol). To avoid conflicts with non-negative, mass-at-zero activations (e.g. ReLU), we state the main parametric approximation at the *pre-activation* level. Define the pre-activation

$$z_i^{(k)}(x,t) := w_i^{(k)}(t)^\top h^{(k-1)}(x,t) + b_i^{(k)}(t), \tag{7}$$

and its smoothed version

$$\tilde{z}_i^{(k)}(x,t) := z_i^{(k)}(x,t) + \varepsilon, \qquad \varepsilon \sim \mathcal{N}(0, \sigma_\varepsilon^2). \tag{8}$$

Diagnostic procedures, variance proxies, and fallback strategies are deferred to the Appendix.

## 3.2 TRAJECTORY-BASED EVOLUTIONARY FITNESS

**Definition 3.4** (Neuron Trajectory). The trajectory of a neuron in state space is defined as

$$\Gamma_i^{(k)} := \{\psi_i^{(k)}(t) \mid t \in [0,T]\}. \tag{9}$$

From this path we extract three complementary quantities:

**Definition 3.5** (Trajectory Length). The trajectory length of neuron $i$ in layer $k$ is the cumulative representational movement of its state vector $\psi_i^{(k)}$ measured with a block-wise scaling matrix $D^{(k)}$ that normalizes heterogeneous components of $\psi$:

$$\mathcal{A}_i^{(k)} := \int_0^T \left\| D^{(k)} \frac{d\psi_i^{(k)}(t)}{dt} \right\|_2 dt, \tag{10}$$

where $D^{(k)}$ is taken to be block-diagonal with positive diagonal blocks that rescale each block of $\psi$. The block-wise construction ensures no single block systematically dominates the norm and makes the quantity invariant to simple coordinate scalings within each block. For comparability across different training durations we use a time-averaged trajectory length; its formal definition and the discrete approximations used in experiments are provided in the Appendix.

**Definition 3.6** (Integrated entropy). The integrated entropy of neuron $i$ accumulates a per-time estimate of the neuron's entropy over training:

$$\mathfrak{H}_i^{(k)} := \int_0^T \mathcal{I}_i^{(k)}(t)\, dt, \tag{11}$$

where $\mathcal{I}_i^{(k)}(t)$ denotes a numerically stable estimator of the neuron's differential entropy at time $t$. For comparability we also consider the time-averaged form $\overline{\mathfrak{H}}_i^{(k)} := \frac{1}{T}\mathfrak{H}_i^{(k)}$.

When the Gaussian plug-in is appropriate (see Assumption 3.3) we use the variance-proxy with explicit numerical stabilization:

$$\tilde{\mathcal{I}}_i^{(k)}(t) := \tfrac{1}{2}\log\big(\operatorname{Var}_x[z_i^{(k)}(x,t)] + \sigma_\varepsilon^2 + \epsilon_{\mathrm{var}}\big), \tag{12}$$

where $\sigma_\varepsilon^2 > 0$ is the additive smoothing noise variance introduced in Assumption 3.3 and $\epsilon_{\mathrm{var}} > 0$ is a small numeric floor (e.g. $10^{-8}$) to avoid $\log(0)$ and ensure robust estimation in finite samples. Note that the Gaussian plug-in differs from the differential entropy by the additive constant $\frac{1}{2}\log(2\pi e)$; when absolute entropy values are needed this constant is accounted for in post-processing.

**Definition 3.7** (Ablation-based utility). For neuron $i$ in layer $k$ define the instantaneous ablation-based utility

$$U_i^{(k)}(t) := \mathbb{E}_{x \sim \mathcal{D}}\big[\mathcal{L}(f_{\theta(t)\setminus i}; x) - \mathcal{L}(f_{\theta(t)}; x)\big], \tag{13}$$

where $f_{\theta\setminus i}$ denotes the network obtained by zeroing neuron $i$'s activation. By this convention $U_i^{(k)}(t) > 0$ indicates the neuron is useful at time $t$.

**Definition 3.8** (Time-averaged utility). For comparability across training durations we use the time-averaged utility

$$\overline{U}_i^{(k)} := \frac{1}{T}\int_0^T U_i^{(k)}(t)\,dt. \tag{14}$$

**Definition 3.9** (Evolutionary fitness). To ensure comparability across heterogeneous quantities we first perform layer-wise standardization (z-scoring) of each constituent statistic and then form a convex combination. Concretely, let

$$\widehat{\overline{U}}_i^{(k)} := \frac{\overline{U}_i^{(k)} - \mathbb{E}_j[\overline{U}_j^{(k)}]}{\mathrm{SD}_j(\overline{U}_j^{(k)})}, \quad \widehat{\overline{\mathcal{S}}}_i^{(k)} := \frac{\overline{\mathcal{S}}_i^{(k)} - \mathbb{E}_j[\overline{\mathcal{S}}_j^{(k)}]}{\mathrm{SD}_j(\overline{\mathcal{S}}_j^{(k)})}, \quad \widehat{\overline{\mathfrak{H}}}_i^{(k)} := \frac{\overline{\mathfrak{H}}_i^{(k)} - \mathbb{E}_j[\overline{\mathfrak{H}}_j^{(k)}]}{\mathrm{SD}_j(\overline{\mathfrak{H}}_j^{(k)})}. \tag{15}$$

The fitness reads

$$\Phi_i^{(k)} := \alpha\,\widehat{\overline{U}}_i^{(k)} - \beta\,\widehat{\overline{\mathcal{S}}}_i^{(k)} + \gamma\,\widehat{\overline{\mathfrak{H}}}_i^{(k)}, \qquad \alpha, \beta, \gamma > 0, \tag{16}$$

where $\alpha, \beta, \gamma$ are either chosen from a small recommended grid after layer-wise normalization or determined by a held-out validation objective. This z-scoring removes unit mismatches and stabilizes comparisons across layers and architectures.

### 3.3 Selection and Survival Criteria

To link fitness to survival, we define thresholds relative to population statistics:

**Definition 3.10** (Survived Neuron). Neuron $i$ in layer $k$ is *survived* if:

$$\Phi_i^{(k)} \geq \mathbb{E}_j[\Phi_j^{(k)}] + \lambda \cdot \mathrm{SD}(\Phi_j^{(k)}), \quad \lambda > 0. \tag{17}$$

This creates an evolutionary-like selection pressure, where only the most informative and stable neurons persist Han (2015); Frankle & Carbin (2019); Morcos (2019).

**Lemma 3.11** (Instability with sustained entropy decay implies vanishing fitness). *Assume there exist constants $c_H > 0$, $T_0 \geq 0$ and $c_S > 0$ such that for all $t \geq T_0$ the neuron's differential entropy satisfies*

$$\mathcal{H}\big(\rho_{i,t}^{(k)}\big) \leq -c_H\,t + C_H, \tag{18}$$

*for some finite constant $C_H$, and furthermore the terminal fluctuation satisfies*

$$\frac{1}{\delta}\int_{t-\delta}^t \Big\|\frac{d\psi_i^{(k)}(s)}{ds}\Big\|_2^2 ds \geq c_S. \tag{19}$$

*Assume also that the time-averaged utility $\overline{U}_i^{(k)}(T)$ and stochasticity $\overline{\mathcal{S}}_i^{(k)}(T)$ grow at most polynomially in $T$. Then for any fixed positive weights $\alpha, \beta, \gamma > 0$ in equation 16 we have*

$$\lim_{T\to\infty} \Phi_i^{(k)}(T) = -\infty. \tag{20}$$

**Definition 3.12** (Gradient–Variance Contribution). For a neuron $i$ in layer $k$ we define the instantaneous gradient second moment

$$q_i^{(k)}(t) := \mathbb{E}_x\Big[\Big(\frac{\partial\mathcal{L}(x)}{\partial a_i^{(k)}(x,t)}\Big)^2\Big], \tag{21}$$

and the instantaneous activation variance

$$\sigma_i^{2(k)}(t) := \mathrm{Var}_x\big[a_i^{(k)}(x,t)\big]. \tag{22}$$

We then define the (time-averaged) *gradient–variance contribution* by

$$\Delta_i^{(k)} := \frac{1}{T}\int_0^T \mathbb{E}_x\Big[\Big(\frac{\partial\mathcal{L}(x)}{\partial a_i^{(k)}(x,t)}\Big)^2 \cdot \mathrm{Var}_x\big[a_i^{(k)}(x,t)\big]\Big]\,dt \tag{23}$$

**Theorem 3.13** (Fitness Threshold Implies Gradient–Variance Contribution). *Let $\Delta_i^{(k)}$ be as above. Suppose Assumptions 3.2 and 3.3 hold, and additionally there exists $\bar{c}_g > 0$ such that the time-averaged gradient second moment satisfies*

$$\frac{1}{T} \int_0^T q_i^{(k)}(t)\, dt \geq \bar{c}_g. \tag{24}$$

*Assume also that $\overline{U}_i^{(k)}(T)$ and $\mathcal{S}_i^{(k)}(T)$ grow at most polynomially in $T$. Then there exist constants $\tau, \kappa > 0$ (depending on $\bar{c}_g, \alpha, \beta, \gamma$ and growth bounds) such that*

$$\Phi_i^{(k)}(T) \geq \tau \quad \Rightarrow \quad \Delta_i^{(k)} \geq \kappa. \tag{25}$$

This result bridges our trajectory-based measure with a classical signal-to-noise criterion, showing that neurons with high fitness necessarily contribute to meaningful gradient–variance interactions Achille & Soatto (2018a); Martens (2020).

Overall, the Neuron Darwinian Dynamics System (NDDS) provides a principled framework to study representational dynamics under Neural Darwinism. Neurons are no longer seen as static units with fixed importance, but as evolving entities competing for survival through their trajectory length, stability, and entropy. This formalism both explains empirical neuron pruning phenomena and predicts inter-layer propagation of specialization Raghu (2017); Jacot (2018).

## 4 EXPERIMENTS

We designed a series of experiments to examine whether CNNs exhibit dynamics consistent with Neural Darwinism, and how such processes shape robustness and representational specialization. Our analysis proceeds in two complementary strands. First, we conduct ablation experiments on a CNN trained on MNIST to quantitatively assess representational resilience under progressive neuron removal. Second, we perform dynamic trajectory analyses across multiple CNN architectures and datasets—ResNet-50 on Tiny-ImageNet—within the framework of the NDDS, with additional experiments on a three-layer MLP-Net with MNIST, ResNet-18 with CIFAR-10, and VGG-16 with CIFAR-100 provided in the Appendix. These experiments share a common methodology—tracking neuron activations, weights, and representational trajectories—while progressively scaling the model depth and dataset complexity. Across all settings, neurons are categorized into survived, eliminated, and other groups based on their long-term representational stability, providing a unified lens for comparing functional contributions across architectures and scales.

### 4.1 ABLATION EXPERIMENT

We conducted ablation experiments using a CNN trained on MNIST to test the resilience of its internal representations under progressive neuron removal. The results are summarized in Figure 1. In the unperturbed network, accuracy reaches 99.3%, and the t-SNE projection reveals tight, well-separated clusters for each digit class, demonstrating a highly structured and linearly separable latent space. When 30% of the neurons are ablated, the accuracy remains essentially unchanged at 99.0%, and the clusters in the t-SNE embedding preserve their compactness and separation, indicating that the representational geometry is only minimally disturbed. This strongly suggests that the network possesses a large degree of representational redundancy. At 60% ablation, accuracy decreases slightly to 98.3%, and the clusters in the t-SNE space begin to expand and partially overlap, particularly at their boundaries. Although separability is degraded, the global structure of the representation is still preserved, implying that the network reallocates representational burden to the remaining subset of neurons. A qualitatively different figure emerges at 90% ablation: accuracy collapses to 64.9%, and the t-SNE projection shows the complete dissolution of the cluster structure, with digit classes intermingled in a disorganized cloud. To summarize, these results provide direct evidence for a Darwinian view of neural representations. Up to moderate levels of ablation, redundant or weakly integrated neurons are eliminated while the core representational structure is maintained, preserving both accuracy and geometric separability. However, once the ablation encroaches upon the Darwinianly selected subset of neurons that are critical for maintaining task-relevant structure, both accuracy and representation quality collapse. This pattern demonstrates that artificial neural networks exhibit precisely the mixture of robustness and selectivity predicted by Neural Darwinism:

multiple neuronal assemblies initially compete to encode overlapping information, but only a small, stabilized ensemble ultimately sustains discriminative capacity under extreme perturbation.

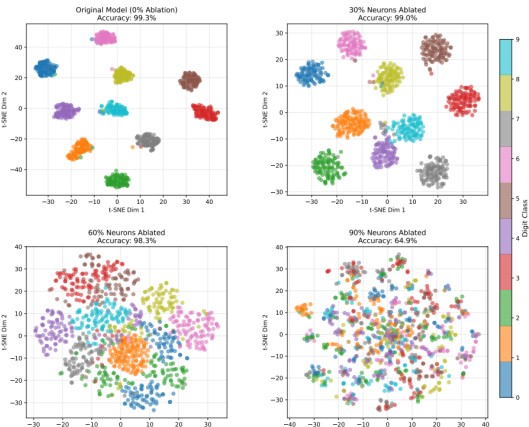

Figure 1: Ablation Experiment on MNIST with Random Neuron Removal.

## 4.2 RESNET-50 ON TINY-IMAGENET

### 4.2.1 DYNAMICS NEURON TRAJECTORY AND EVOLUTION ANALYSIS

The dynamic PCA trajectories for the shallow layer (Figure 2(a), top) provide a temporal view of representational changes across training. Each trajectory reflects the evolution of a neuron's activation statistics in a low-dimensional PCA space. Survived neurons generally trace longer and more directionally consistent paths; this pattern is consistent with representational refinement and greater task-related adaptation. These trajectories tend to drift toward more structured regions of the PCA manifold, indicating a non-random reorganization that supports discriminative feature encoding. By contrast, eliminated neurons follow noticeably shorter, less exploratory trajectories that remain close to their initial locations in PCA space. This limited movement is consistent with functional stagnation in the sense of limited representational development. Such stagnation is consistent with patterns one might expect in early-stage selective pruning (i.e., neurons with limited representational change tend to be removed over training). Quantitative analysis reinforces these patterns. By the final epoch (Figure 2(c), top), survived neurons reach a median cumulative trajectory length of approximately 3.2 units, compared to 2.4 for eliminated neurons and around 2.3 for the other group. These results indicate an association between sustained representational movement (rather than initial position) and retention. Weight magnitude evolution (Figure 2(d), top) shows only minor differences across groups: eliminated neurons maintain slightly higher L2 norms than survived, with other neurons consistently lowest. The overall stability across training suggests that in shallow layers, synaptic resource allocation is relatively stable, with large-scale reallocation not yet evident.

The PCA trajectories for the middle layer (Figure 2(a), middle) capture a more pronounced divergence in representational dynamics across neuron types. Survived neurons traverse extended, often curved paths in the PCA space, largely oriented along PC1 (96.7% variance explained), with modest modulation along PC2 (3.1%). Although some trajectories exhibit partial rightward drift, clustering is weak and dispersion remains the dominant pattern. Eliminated neurons show substantially shorter displacements, remaining near their initialization points with fragmented paths. The intermediate other group exhibits moderate movement but does not match the sustained displacement of survivors. Trajectory length evolution (Figure 2(c), middle)) highlights this separation: by the end of training, survived neurons reach approximately 3.8 cumulative units, while eliminated neurons plateau near 2.8, with the other group is even lower. The gap is wider than in the shallow layer, underscoring that sustained representational plasticity becomes increasingly decisive at mid-level processing stages. Weight magnitude evolution (Figure 2(d), middle)) shows relatively stable rankings: eliminated neurons hold slightly higher norms than survived. The lack of pronounced growth for eliminated neurons—despite higher absolute values—suggests that strong initial parameterization was not matched by functional adaptation.

The dynamic PCA trajectories for the deep layer (Figure 2(a), bottom)) reveal the strongest differentiation in representational mobility. Survived neurons navigate long, structured arcs, reflecting continued refinement and consolidation of high-level semantic representations. These trajectories exhibit a clear convergence trend toward a more compact subregion of the PCA manifold, consistent with the emergence of attractor-like states that dominate the network's final decision space. Eliminated neurons, in contrast, show markedly shorter trajectories, with minimal displacement beyond early training epochs, indicating rapid stagnation. Other neurons display partial mobility but fail to achieve the sustained, directional movement observed in survivors. Trajectory length analysis (Figure 2(c), bottom)) accentuates this contrast: by the final epoch, survived neurons reach 7 cumulative units, while eliminated neurons remain near 4. This substantial gap shows that greater representational plasticity is strongly associated with deep-layer survival. Weight magnitude evolution (Figure 2(d), bottom) exhibit a global decay across all neuron types, converging toward lower norms over training. Survived and eliminated neurons follow similar L2 trajectories with only slight divergence at convergence, while the other group tends toward lower values. These patterns are consistent with reduced differentiation of synaptic strength in deeper layers and indicate that survival correlates with only marginally higher residual weights. Overall, these findings are consistent with a progressively stronger association between our measured dynamics and neuron retention with increasing depth. In shallow layers, selection pressure is relatively permissive, with only subtle differences in trajectory and weight dynamics. In middle layers, divergence intensifies, as sustained plasticity becomes a critical factor for survival. In deep layers, we observe patterns consistent with consolidation—neurons that exhibit larger representational changes are more likely to be retained and may contribute disproportionately to high-level representations. These results are broadly consistent with components of the Neural Darwinism framework—variation, competition, and selective retention—insofar as our measures show compatible patterns.

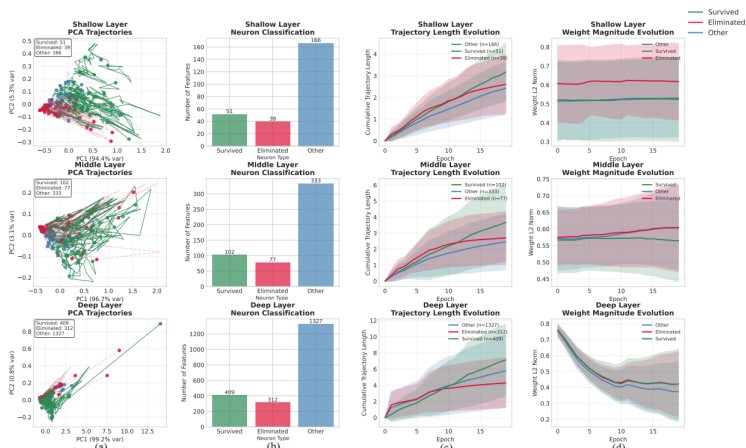

Figure 2: Dynamics Neuron Trajectory and Evolution Analysis on Tiny-ImageNet.

### 4.2.2 STATIC PCA AND ACTIVATION EVOLUTION

Figure 3 presents static PCA projections of final neuron states (top row) and mean activation norm trajectories (bottom row) across shallow, middle, and deep layer. In the shallow layer PC1 explains 94.4% of the variance (PC2 5.3%), which suggests that the final neuron population is largely confined to a single dominant axis in the projected space. Survived neurons (green) occupy a moderately dispersed region displaced from the origin, consistent with coordinated stabilization that does not form a tightly compact cluster. Eliminated neurons (red) form a compact cluster near the lower-left quadrant; this spatial concentration is consistent with lower mean activation magnitude. Other neurons (blue) lie in an intermediate zone, reflecting partial but incomplete adaptation. Activation dynamics are consistent with the PCA structure: on average survived neurons exhibit higher and more stable norms, eliminated neurons show a downward trend toward near-zero activity, and other neurons follow an intermediate trajectory.

In the middle layer PC1 accounts for 96.7% of variance (PC2 3.1%), indicating a stronger alignment to a single dominant direction compared to the shallow layer. Neurons distribute primarily along

this axis: survivors occupy the central and positive range of PC1, reflecting sustained functional activity; eliminated neurons cluster near the negative end of PC1, marking progressive silencing; and other neurons lie in between. Activation dynamics mirror this structure: survivors maintain consistently higher norms, eliminated neurons decay rapidly toward inactivity, and others exhibit moderate decline. These patterns are consistent with an increasingly directional consolidation in the middle layer: survivors tend to concentrate along the principal subspace while eliminated neurons are displaced toward the opposite pole.

In the deep layer PC1 captures 99.2% of the variance (PC2 0.8%), suggesting that neuron states are largely ordered along a single dominant axis in the projected space. Neurons concentrate into a dense central region dominated by other units; eliminated neurons tend to localize near the low-PC1 boundary while survived neurons extend toward the positive-PC1 tail. Activation trajectories are consistent with this separation: survivors typically increase early in training and then stabilize at higher mean norms, eliminated neurons decline rapidly toward near-zero on average, and other units tend to plateau at intermediate values. These dynamics are consistent with an axis-aligned selection process in which survival status correlates with displacement along the dominant representational axis. Taken together, the layerwise progression is consistent with selection-like dynamics: initial heterogeneity, a preferential decline of low-activity units, and selective retention of survivors that increasingly align with task-relevant representational axes. The increasing dominance of a single principal axis and the widening separation in activation dynamics are consistent with a layerwise intensification of selective pressures, culminating in increased specialization in deeper layers.

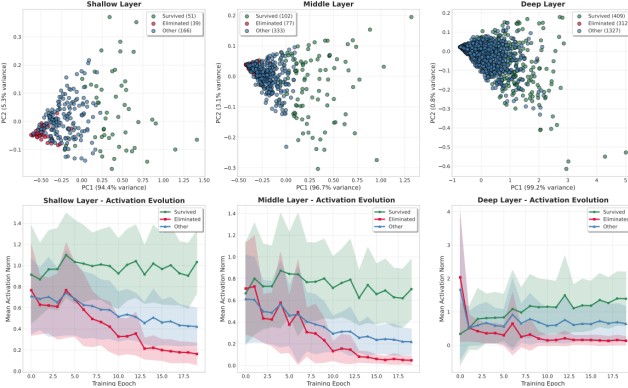

Figure 3: Static PCA and Activation Evolution on Tiny-ImageNet.

## 5 CONCLUSION

This study provides empirical evidence that CNNs exhibit representational dynamics that are consistent with the principles of Neural Darwinism. Across architectures and datasets, we observe recurring signatures of variation, competition, and selective retention: neurons initially follow diverse representational trajectories, but only a subset sustains adaptive movement, stronger weight magnitudes, and higher activation norms. The ablation experiment highlights both robustness, arising from representational redundancy, and fragility, once the implicitly selected subset of critical neurons is disrupted. Layerwise analyses further suggest that selection pressure intensifies with depth, culminating in compact ensembles of specialized neurons that dominate high-level feature encoding.

These findings advance our understanding of representation learning by framing it not solely as gradient-driven optimization, but also as an emergent selection-like process operating at the neuron level. This dual perspective highlights how neural networks balance redundancy with specialization. Future work may investigate whether similar dynamics generalize to recurrent and transformer architectures, and explore implications for pruning, interpretability, and biologically inspired models of computation. [1]

---

[1] We used large language models (LLMs) only for polishing the writing; all scientific content is the authors' own.

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

# A  APPENDIX

## A.1  NOTATION AND PRELIMINARIES

To maintain consistency with the main text, we briefly recap key notations:

- Neural network $f_\theta : \mathcal{X} \to \mathcal{Y}$, layers $\{L_k\}_{k=1}^D$, where layer $k$ contains $n_k$ neurons indexed by $i$.

- Parameters of neuron $i$ at layer $k$: weights $w_i^{(k)}(t) \in \mathbb{R}^{d_{k-1}}$, bias $b_i^{(k)}(t) \in \mathbb{R}$, activation function $\sigma$.

- Activation:
$$a_i^{(k)}(x,t) := \sigma\left(\langle w_i^{(k)}(t), h^{(k-1)}(x,t)\rangle + b_i^{(k)}(t)\right). \tag{26}$$

- Neuron state vector (compound state):
$$\psi_i^{(k)}(t) := \left[w_i^{(k)}(t),\, b_i^{(k)}(t),\, \mu_i^{(k)}(t),\, g_i^{(k)}(t),\, \mathcal{I}_i^{(k)}(t)\right], \tag{27}$$

where
$$\mu_i^{(k)}(t) = \mathbb{E}_{x\sim\mathcal{D}}[a_i^{(k)}(x,t)], \quad g_i^{(k)}(t) = \mathbb{E}_{x\sim\mathcal{D}}\left[\frac{\partial \mathcal{L}(x)}{\partial a_i^{(k)}(x,t)}\right], \tag{28}$$

and $\mathcal{I}_i^{(k)}(t)$ is the instantaneous differential (Shannon) entropy estimator of the activation distribution. The integrated (accumulated) entropy over training is denoted $\mathfrak{H}_i^{(k)}$ as in the main text.

- State evolution (ODE form, main text eq.(6)):
$$\frac{d}{dt}\psi_i^{(k)}(t) = \mathbf{F}_\theta^{(k)}\left(\psi_i^{(k)}(t), \mathcal{D}, \mathcal{L}\right). \tag{29}$$

Other quantities such as trajectory length $\mathcal{A}_i^{(k)}$, terminal stochasticity $\mathcal{S}_i^{(k)}$, integrated entropy $\mathfrak{H}_i^{(k)}$, and fitness $\Phi_i^{(k)}$ follow the main text definitions. **Notation remark:** throughout the manuscript we reserve $\mathcal{L}(\cdot)$ exclusively for the per-example loss; the trajectory length is consistently denoted $\mathcal{A}_i^{(k)}$.

## A.2  SUPPLEMENTARY TECHNICAL ASSUMPTIONS

We explicitly state additional mild assumptions needed for mathematical rigor and numerical stability. These assumptions clarify the hidden conditions of the main results.

**Assumption S1 (Smoothness, boundedness, and trajectory length)**

For each layer $k$, the vector field $\mathbf{F}_\theta^{(k)}(\psi, t)$ is locally Lipschitz in $\psi$ and measurable in $t$. There exist constants $B_g, B_a, B_\psi > 0$ such that for all $t \geq 0$:

$$\|g_i^{(k)}(t)\| \leq B_g, \quad \mathrm{Var}[a_i^{(k)}(t)] \leq B_a, \quad \|\psi_i^{(k)}(t)\| \leq B_\psi. \tag{30}$$

Moreover, the trajectory (arc) length $\mathcal{A}_i^{(k)}(T)$ is bounded for any finite $T$.

**Assumption S2 (Sub-exponential tails / sub-Gaussianity of activations)**

For all neurons $i, k$ and times $t$, the distribution of $a_i^{(k)}(x,t)$ over $x \sim \mathcal{D}$ is sub-Gaussian or at least has sub-exponential tails, enabling concentration bounds for sample estimators.

**Assumption S3 (Controlled Gaussian entropy approximation error)**

There exists a constant $C_{\mathrm{gauss}} \geq 1$ such that for all neurons $i, k$ and times $t$,

$$\mathcal{I}_i^{(k)}(t) \leq \frac{1}{2}\log\left(2\pi e\,\mathrm{Var}[a_i^{(k)}(t)]\right) \leq \mathcal{I}_i^{(k)}(t) + \log C_{\mathrm{gauss}}. \tag{31}$$

This controlled approximation underpins the Gaussian plug-in used in experiments; when this bound is violated the practitioner must rely on nonparametric estimators as described in the main text.

### A.3 Well-Posedness of the Continuous NDDS

Under Assumption S1, the vector field $\mathbf{F}_\theta^{(k)}$ is locally Lipschitz, thus by Picard–Lindelöf theorem Shih (2023); Yarotsky (2024), for any initial value $\psi_i^{(k)}(0)$ there exists a unique local solution. Boundedness and growth controls ensure global existence on finite intervals and continuous dependence on initial conditions and parameters.

### A.4 Smooth and Bounded Dynamics

**Assumption A.1** (Smooth and Bounded Dynamics). We adopt a hypothesis compatible with practical discrete optimization. The parameter trajectory $\theta(t)$ is assumed to be absolutely continuous and piecewise $C^1$ in $t$ (so that it admits a time-continuous interpolation), and $\mathbf{F}_\theta^{(k)}$ is locally Lipschitz in $\psi$ on trajectories of interest. This formulation explicitly permits discretization effects arising from SGD and non-smooth activations (e.g. ReLU) by interpreting derivatives in the sense of absolutely continuous interpolation or Clarke subgradients when necessary.

Furthermore, there exist constants $B_g, B_a > 0$ such that for all $t \in [0, T]$ along the interpolated trajectory:

$$\|g_i^{(k)}(t)\| \le B_g, \quad \mathrm{Var}[a_i^{(k)}(t)] \le B_a. \tag{32}$$

Finally, we require that the trajectory length $\mathcal{A}_i^{(k)}$ (defined in equation 49) remains finite as $T \to \infty$; for discrete checkpoints the forward-difference approximation in equation 51 is used and all continuum claims are understood to hold up to discretization errors that vanish under standard time-interpolation refinements.

#### A.4.1 Discrete Continuous Trajectory Length Approximation

**Setup.** Let $a : [0, T] \to \mathbb{R}^d$ be the neuron activation trajectory $a(t) \equiv a_i^{(k)}(t)$ appearing in Assumption "Smooth and Bounded Dynamics". Assume $a$ is absolutely continuous on $[0, T]$ (hence a.e. differentiable with $a' \in L^1([0, T]; \mathbb{R}^d)$) and has finite Trajectory length

$$\mathcal{A} = \int_0^T \|a'(t)\| \, dt < \infty. \tag{33}$$

For a uniform partition $0 = t_0 < t_1 < \cdots < t_M = T$ with step size $\Delta t = T/M$ define the forward-difference (discrete) trajectory length approximation

$$\widehat{\mathcal{A}}(\Delta t) = \sum_{m=1}^M \|a(t_m) - a(t_{m-1})\| = \sum_{m=1}^M \Big\| \int_{t_{m-1}}^{t_m} a'(s) \, ds \Big\|. \tag{34}$$

**Lemma A.2** (Discrete Continuous Trajectory Length Approximation). *Under the setup above the following hold.*

1. *Convergence. As the mesh $\Delta t \to 0$,*

$$\widehat{\mathcal{A}}(\Delta t) \longrightarrow \mathcal{A}. \tag{35}$$

*In particular, for any sequence of partitions whose mesh size tends to zero the partition-wise variation of $a$ converges to the total variation (trajectory length) $\mathcal{A}$.*

2. *Quantitative bound under extra smoothness. If, in addition, $a'$ is $L$-Lipschitz on $[0, T]$ (i.e. there exists $L > 0$ such that $\|a'(s) - a'(t)\| \le L|s - t|$ for all $s, t \in [0, T]$), then there exists a constant $C$ (one may take $C = L$) such that for all sufficiently small $\Delta t$:*

$$\big| \mathcal{A} - \widehat{\mathcal{A}}(\Delta t) \big| \le C T \Delta t = O(\Delta t). \tag{36}$$

3. *Non-smooth activations (Clarke subgradient). If $a$ is only piecewise $C^1$ (for example due to ReLU kinks) and is absolutely continuous, interpret $a'$ in the Clarke subdifferential sense. Then the convergence in part (1) still holds; moreover, whenever the extra smoothness of part (2) holds on each $C^1$ segment the $O(\Delta t)$ bound applies up to contributions from finitely many kink-boundary intervals, which vanish as $\Delta t \to 0$.*

*Proof.* **(1) Convergence.** Absolute continuity of $a$ implies $a$ has bounded variation on $[0, T]$ and

$$\mathcal{A} = \text{Var}(a; [0, T]) = \sup_{\mathcal{P}} \sum_{m} \|a(t_m) - a(t_{m-1})\|, \tag{37}$$

where the supremum is taken over all finite partitions $\mathcal{P}$ of $[0, T]$. For any fixed partition the sum $\sum_m \|a(t_m) - a(t_{m-1})\|$ is the variation of $a$ over that partition and is therefore bounded above by $\mathcal{A}$. Standard results on functions of bounded variation state that for any sequence of partitions whose mesh tends to zero the corresponding partition-wise variation converges to the total variation. Applied to the uniform partitions above this yields

$$\lim_{\Delta t \to 0} \widehat{\mathcal{A}}(\Delta t) = \mathcal{A}, \tag{38}$$

which proves (1).

**(2) Quantitative bound under Lipschitz derivative.** Assume $a'$ is $L$-Lipschitz Havens (2023); Bertrand (2024). Fix an interval $I_m = [t_{m-1}, t_m]$. By the fundamental theorem of calculus and the Lipschitz property we can expand $a'$ about the midpoint (or any point $\xi_m \in I_m$) to obtain

$$\int_{t_{m-1}}^{t_m} a'(s)\, ds = \Delta t\, a'(\xi_m) + r_m, \tag{39}$$

with the remainder satisfying $\|r_m\| \leq \frac{1}{2} L (\Delta t)^2$. Hence

$$\left\| \int_{t_{m-1}}^{t_m} a'(s)\, ds \right\| = \Delta t \|a'(\xi_m)\| + \delta_m, \qquad |\delta_m| \leq \frac{1}{2} L (\Delta t)^2. \tag{40}$$

On the other hand,

$$\int_{t_{m-1}}^{t_m} \|a'(s)\|\, ds = \Delta t \|a'(\xi_m)\| + \epsilon_m, \qquad |\epsilon_m| \leq L (\Delta t)^2, \tag{41}$$

where the bound on $\epsilon_m$ follows from the same Lipschitz control on $a'$ and the one-dimensional integral averaging error. Subtracting and summing over $m = 1, \dots, M$ yields

$$0 \leq \mathcal{A} - \widehat{\mathcal{A}}(\Delta t) = \sum_{m=1}^{M} \left( \int_{t_{m-1}}^{t_m} \|a'(s)\|\, ds - \left\| \int_{t_{m-1}}^{t_m} a'(s)\, ds \right\| \right) \leq \sum_{m=1}^{M} \left( |\epsilon_m| + |\delta_m| \right). \tag{42}$$

Using the per-interval bounds $|\epsilon_m| \leq L (\Delta t)^2$, $|\delta_m| \leq \frac{1}{2} L (\Delta t)^2$ we obtain

$$\left| \mathcal{A} - \widehat{\mathcal{A}}(\Delta t) \right| \leq \frac{3}{2} L M (\Delta t)^2 = \frac{3}{2} L T \Delta t. \tag{43}$$

Thus the difference is $O(\Delta t)$; setting $C = \frac{3}{2} L$ (or taking the coarser but simpler $C = L$) yields the claimed linear-in-$\Delta t$ bound.

**(3) Non-smooth activations and Clarke subgradient.** If $a$ is piecewise $C^1$ (typical when activations like ReLU produce kinks) then $a$ is still absolutely continuous and has bounded variation. The set $K \subset [0, T]$ of non-differentiable points is closed and of Lebesgue measure zero (in common architectures it is finite or a countable set with no accumulation inside $[0, T]$). The contribution of intervals that contain points of $K$ can be localized: by refining the partition one can make the total length of intervals that intersect $K$ arbitrarily small, hence their contribution to $\mathcal{A}$ and to the discrete sum is arbitrarily small. On each $C^1$ segment the argument of part (2) applies; summing segment-wise yields the same $O(\Delta t)$ behaviour up to vanishing boundary contributions. More conceptually, one may replace $a'$ by any measurable selection from the Clarke generalized derivative Park (2024) and repeat the preceding estimates; the measure-zero nondifferentiable set does not affect the limiting equality $\widehat{\mathcal{A}}(\Delta t) \to \mathcal{A}$ nor the $O(\Delta t)$ rate when the Lipschitz condition holds on the smooth pieces.

**Remark.** In typical empirical settings the checkpoint count $M$ is large (e.g. hundreds or thousands), so $\Delta t = T/M$ is small and the discretization error $|\mathcal{A} - \widehat{\mathcal{A}}(\Delta t)|$ is negligible compared to stochastic fluctuations induced by SGD. The theoretical statements above make precise that all continuous-time claims involving $\mathcal{A}$ hold up to an $O(\Delta t)$ discretization error which vanishes under standard time-interpolation refinements. $\qquad \square$

## A.5 LOCAL GAUSSIANITY OF PRE-ACTIVATIONS AND DIAGNOSTIC PROTOCOL

**Assumption A.3** (Local Gaussianity of pre-activations and diagnostic protocol). To avoid conflicts with non-negative, mass-at-zero activations (e.g. ReLU), we state the main parametric approximation at the *pre-activation* level. Define the pre-activation

$$z_i^{(k)}(x, t) := w_i^{(k)}(t)^\top h^{(k-1)}(x, t) + b_i^{(k)}(t), \tag{44}$$

and its smoothed version

$$\tilde{z}_i^{(k)}(x, t) := z_i^{(k)}(x, t) + \varepsilon, \qquad \varepsilon \sim \mathcal{N}(0, \sigma_\varepsilon^2). \tag{45}$$

For every neuron $i$ and for any short time window $[s, s+\tau]$ (with $\tau$ chosen to balance local stationarity and sample requirements) we assume that the empirical law of $\tilde{z}_i^{(k)}(\cdot, s)$ is well-approximated by a Gaussian $\mathcal{N}(\mu_{z,i}^{(k)}(s), \sigma_{z,i}^{2(k)}(s))$ in the sense that there exists a small tolerance $\eta > 0$ and a divergence metric $\mathrm{dist}(\cdot, \cdot)$ such that for a large fraction of checkpoints $s \in [0, T]$,

$$\mathrm{dist}\big(\mathrm{Law}(\tilde{z}_i^{(k)}(\cdot, s)), \, \mathcal{N}(\mu_{z,i}^{(k)}(s), \sigma_{z,i}^{2(k)}(s))\big) \leq \eta. \tag{46}$$

When downstream analysis requires activation-level entropy (post-activation), practitioners must either transform the Gaussian approximation via the known mapping $\sigma(\cdot)$ and report the accuracy of that transformation, or employ a consistent nonparametric estimator for the activation distribution and report estimator sensitivity.

A numerically-stable variance-proxy is used when the pre-activation Gaussian plug-in is accepted:

$$\tilde{\mathcal{I}}_i^{(k)}(t) := \tfrac{1}{2} \log\big( \mathrm{Var}_x[z_i^{(k)}(x, t)] + \sigma_\varepsilon^2 + \epsilon_{\mathrm{var}}\big). \tag{47}$$

If the Gaussian diagnostic fails (i.e. the empirical divergence exceeds $\eta$) the practitioner must fall back to nonparametric estimators and report the fraction of checkpoints failing the diagnostic and a sensitivity comparison between plug-in and nonparametric estimates.

## A.6 NEURON TRAJECTORY

**Definition A.4** (Neuron Trajectory). The trajectory of a neuron in state space is defined as

$$\Gamma_i^{(k)} := \{\psi_i^{(k)}(t) \mid t \in [0, T]\}. \tag{48}$$

From this path we extract three complementary quantities:

**Definition A.5** (Trajectory Length). The trajectory length of neuron $i$ in layer $k$ is the cumulative representational movement of its state vector $\psi_i^{(k)}$ measured with a block-wise scaling matrix $D^{(k)}$ that normalizes heterogeneous components of $\psi$:

$$\mathcal{A}_i^{(k)} := \int_0^T \left\| D^{(k)} \frac{d\psi_i^{(k)}(t)}{dt} \right\|_2 dt, \tag{49}$$

where $D^{(k)}$ is taken to be block-diagonal with positive diagonal blocks that rescale each block of $\psi$. The block-wise construction ensures no single block systematically dominates the norm and makes the quantity invariant to simple coordinate scalings within each block.

When comparability across different training durations is required we also use the time-averaged arc length

$$\overline{\mathcal{A}}_i^{(k)} := \frac{1}{T} \mathcal{A}_i^{(k)}. \tag{50}$$

Under discrete training (checkpoints or optimization steps with index spacing $\Delta t$) we employ the forward-difference approximation

$$\mathcal{A}_i^{(k)} \approx \sum_{t=0}^{N-1} \left\| D^{(k)} \frac{\psi_i^{(k)}(t+1) - \psi_i^{(k)}(t)}{\Delta t} \right\|_2 \Delta t, \tag{51}$$

where $N$ is the number of recorded checkpoints and $\Delta t$ is the (possibly non-unit) interval between checkpoints; taking $\Delta t = 1$ recovers the step-indexed form.

## A.7 INTEGRATED ENTROPY

**Definition A.6** (Integrated entropy). The integrated entropy of neuron $i$ accumulates a per-time estimate of the neuron's (differential) entropy over training:

$$\mathfrak{H}_i^{(k)} := \int_0^T \mathcal{I}_i^{(k)}(t)\, dt, \tag{52}$$

where $\mathcal{I}_i^{(k)}(t)$ denotes a numerically stable estimator of the neuron's differential entropy at time $t$ (estimated from mini-batches and moving averages). For comparability we also consider the time-averaged form $\overline{\mathfrak{H}}_i^{(k)} := \frac{1}{T}\mathfrak{H}_i^{(k)}$.

When the Gaussian plug-in is appropriate (see Assumption 3.3) we use the variance-proxy with explicit numerical stabilization:

$$\tilde{\mathcal{I}}_i^{(k)}(t) := \tfrac{1}{2}\log\big(\operatorname{Var}_x[z_i^{(k)}(x,t)] + \sigma_\varepsilon^2 + \epsilon_{\text{var}}\big), \tag{53}$$

where $\sigma_\varepsilon^2 > 0$ is the additive smoothing noise variance introduced in Assumption 3.3 and $\epsilon_{\text{var}} > 0$ is a small numeric floor (e.g. $10^{-8}$) to avoid $\log(0)$ and ensure robust estimation in finite samples. Note that the Gaussian plug-in differs from the differential entropy by the additive constant $\frac{1}{2}\log(2\pi e)$; when absolute entropy values are needed this constant is accounted for in post-processing.

In discrete form the accumulated entropy used in experiments is

$$\mathfrak{H}_i^{(k)} \approx \sum_{t=0}^{N-1} \tilde{\mathcal{I}}_i^{(k)}(t)\, \Delta t, \tag{54}$$

with $\Delta t$ equal to the checkpoint interval. When the Gaussian assumption is questionable (e.g. ReLU activations with large mass at zero), we complement the variance-proxy with nonparametric estimators. Estimation uses mini-batch averages with an exponential moving-average smoothing window.

## A.8 PRACTICAL ESTIMATOR FOR ABLATION-BASED UTILITY

**Definition A.7** (Ablation-based utility). For neuron $i$ in layer $k$ define the instantaneous ablation-based utility

$$U_i^{(k)}(t) := \mathbb{E}_{x\sim\mathcal{D}}\big[\mathcal{L}(f_{\theta(t)\backslash i}; x) - \mathcal{L}(f_{\theta(t)}; x)\big], \tag{55}$$

where $f_{\theta\backslash i}$ denotes the network obtained by zeroing neuron $i$'s activation. By this convention $U_i^{(k)}(t) > 0$ indicates the neuron is useful at time $t$.

Direct computation of equation 55 for every neuron at every checkpoint is computationally prohibitive. We therefore recommend and use a calibrated first-order Taylor approximation Garibbo (2023); Sun (2023) as a default estimator (and validate it against ground-truth partial ablations on small models):

$$U_i^{(k)}(t) \approx -\mathbb{E}_x\left[\frac{\partial\mathcal{L}(x)}{\partial a_i^{(k)}(x,t)} \cdot a_i^{(k)}(x,t)\right] =: U_i^{(k),\text{lin}}(t). \tag{56}$$

Optionally, a second-order correction may be included when Hessian-vector products are affordable. In practice we compute $U_i^{(k),\text{lin}}(t)$ using a held-out validation subset of size $m \ll |\mathcal{D}|$ (randomly sampled) and report the estimator variance and a small-sample calibration against exact ablation on a subset of neurons.

## A.9 DETAILED PROOFS OF MAIN LEMMAS AND THEOREMS

**Lemma A.8** (Instability with sustained entropy decay implies vanishing fitness). *Assume there exist constants $c_H > 0$, $T_0 \geq 0$ and $c_S > 0$ such that for all $t \geq T_0$ the neuron's differential entropy satisfies*

$$\mathcal{H}\big(\rho_{i,t}^{(k)}\big) \leq -c_H\, t + C_H, \tag{57}$$

*for some finite constant $C_H$, and furthermore the terminal fluctuation satisfies*

$$\frac{1}{\delta}\int_{t-\delta}^t \left\|\frac{d\psi_i^{(k)}(s)}{ds}\right\|_2^2 ds \geq c_S. \tag{58}$$

*Assume also that the time-averaged utility $\overline{U}_i^{(k)}(T)$ and stochasticity $\overline{S}_i^{(k)}(T)$ grow at most polynomially in $T$. Then for any fixed positive weights $\alpha, \beta, \gamma > 0$ in equation 16 we have*

$$\lim_{T\to\infty} \Phi_i^{(k)}(T) = -\infty. \tag{59}$$

*Proof.* A linear-in-time *growth* of the relative entropy $\mathrm{KL}(\rho_{i,t}^{(k)} \| \rho_{\mathrm{ref}})$ implies that the neuron's differential entropy (and hence the Gaussian plug-in proxy used in $\mathfrak{H}_i^{(k)}$) decreases sufficiently fast. After the layer-wise standardization in equation 16, this persistent loss of information eventually dominates the (assumed at-most-polynomial) contributions from $\overline{U}$ and $\overline{S}$, driving $\Phi_i^{(k)}(T) \to -\infty$. $\square$

**Theorem A.9** (Fitness Threshold Implies Gradient–Variance Contribution). *Let $\Delta_i^{(k)}$ be as above. Suppose Assumptions 3.2 and 3.3 hold, and additionally there exists $\bar{c}_g > 0$ such that the time-averaged gradient second moment satisfies*

$$\frac{1}{T} \int_0^T q_i^{(k)}(t)\, dt \geq \bar{c}_g. \tag{60}$$

*Assume also that $\overline{U}_i^{(k)}(T)$ and $S_i^{(k)}(T)$ grow at most polynomially in $T$. Then there exist constants $\tau, \kappa > 0$ (depending on $\bar{c}_g, \alpha, \beta, \gamma$ and growth bounds) such that*

$$\Phi_i^{(k)}(T) \geq \tau \quad \Rightarrow \quad \Delta_i^{(k)} \geq \kappa. \tag{61}$$

*Proof.* Under Assumption 3.3 the time-averaged pre-activation variance $\overline{\sigma^2} := \frac{1}{T} \int_0^T \mathrm{Var}_x[z_i^{(k)}(x,t)]\, dt$ is related to $\mathfrak{H}_i^{(k)}$ via the Gaussian plug-in. After the layer-wise z-scoring used in equation 16 a lower bound on $\Phi$ yields a lower bound on $\overline{\sigma^2}$ up to contributions from $\mathcal{L}$ and $\mathcal{S}$. Combining this with the time-averaged lower bound on $q_i^{(k)}$ gives

$$\Delta_i^{(k)} \geq \bar{c}_g \cdot \overline{\sigma^2}, \tag{62}$$

and the constants $\tau, \kappa$ follow by quantitative bookkeeping of the contributions of $\mathcal{L}$ and $\mathcal{S}$. $\square$

## A.10 DISCRETE-TIME APPROXIMATION AND RELATION TO SGD

Actual training proceeds in discrete time steps, typically iterations or epochs. The continuous-time NDDS dynamics approximate the discrete SGD updates as follows:

- Discrete parameter update:

$$\theta_{t+1} = \theta_t - \eta_t \widehat{\nabla}_\theta \mathcal{L}(B_t; \theta_t), \tag{63}$$

  where $B_t$ is the mini-batch at step $t$.
- For small learning rate $\eta_t$, the discrete updates approximate the stochastic differential equation

$$d\theta_t = -\mathbb{E}_x[\nabla_\theta \mathcal{L}(x; \theta_t)]dt + \sqrt{\eta_t}\Sigma(\theta_t)dW_t, \tag{64}$$

  with $W_t$ Brownian motion and $\Sigma$ the noise covariance.
- Correspondingly, the neuron state differences

$$\Delta\psi_i^{(k)}(t) := \psi_i^{(k)}(t+1) - \psi_i^{(k)}(t) \tag{65}$$

  approximate $\frac{d}{dt}\psi_i^{(k)}(t)$.
- Therefore,

$$\mathcal{A}_i^{(k)} \approx \sum_t \|\Delta\psi_i^{(k)}(t)\|_2, \quad \mathcal{S}_i^{(k)} \approx \frac{1}{\delta} \sum_{t=T-\delta}^{T-1} \|\Delta\psi_i^{(k)}(t)\|_2^2, \quad \mathfrak{H}_i^{(k)} \approx \sum_t \mathcal{I}_i^{(k)}(t). \tag{66}$$

Discrete estimation errors arise from step size, mini-batch noise, and finite sample effects. In all discrete approximations used in experiments we adopt the same block-wise scaling matrix $D^{(k)}$ that appears in the continuous trajectory length definition (main text Eq. equation 49) to ensure consistent units across measurements.

## A.11 NUMERICAL ESTIMATION OF KEY QUANTITIES

**Definition A.10** (Mean activation and mean gradient). Given an evaluation dataset $\mathcal{D}_{\mathrm{eval}}$, the mean activation and mean gradient of neuron $i$ in layer $k$ are estimated as

$$\mu_i^{(k)} = \frac{1}{|\mathcal{D}_{\mathrm{eval}}|} \sum_{x \in \mathcal{D}_{\mathrm{eval}}} a_i^{(k)}(x), \quad g_i^{(k)} = \frac{1}{|\mathcal{D}_{\mathrm{eval}}|} \sum_{x \in \mathcal{D}_{\mathrm{eval}}} \frac{\partial \mathcal{L}(x)}{\partial a_i^{(k)}(x)}. \tag{67}$$

**Definition A.11** (Activation variance). The variance of activations is estimated as the unbiased sample variance over $\mathcal{D}_{\mathrm{eval}}$:

$$\widehat{\mathrm{Var}}[a_i^{(k)}] = \frac{1}{|\mathcal{D}_{\mathrm{eval}}| - 1} \sum_{x \in \mathcal{D}_{\mathrm{eval}}} \left( a_i^{(k)}(x) - \mu_i^{(k)} \right)^2. \tag{68}$$

**Definition A.12** (Differential entropy). We consider three standard estimators for the entropy of activations:

1. **Gaussian plug-in:**
$$\widehat{\mathcal{I}}_{\mathrm{gauss}} = \tfrac{1}{2} \log\left( 2\pi e \, \widehat{\mathrm{Var}}[a_i^{(k)}] \right), \tag{69}$$
    with a numeric floor $\epsilon_{\mathrm{var}} > 0$ (Eq. equation 53) to avoid degeneracy.

2. **Kernel density estimation (KDE):** Estimate density $\widehat{p}(z)$ via KDE and compute
$$\widehat{\mathcal{I}} = -\int \widehat{p}(z) \log \widehat{p}(z) \, dz. \tag{70}$$

3. **K-nearest neighbor (Kozachenko–Leonenko):** Nonparametric entropy estimation based on neighbor distances.

**Definition A.13** (Trajectory length and terminal stochasticity). From saved parameter snapshots at discrete steps $t$, define the scaled increment

$$\Delta\psi_i^{(k)}(t) := \left\| D^{(k)}\big(\psi_i^{(k)}(t+1) - \psi_i^{(k)}(t)\big) \right\|_2, \tag{71}$$

where $D^{(k)}$ is the block-wise scaling matrix. Then the trajectory length and terminal stochasticity are given by

$$\mathcal{A}_i^{(k)} = \sum_t \Delta\psi_i^{(k)}(t), \quad \mathcal{S}_i^{(k)} = \frac{1}{\delta} \sum_{t=T-\delta}^{T-1} \left( \Delta\psi_i^{(k)}(t) \right)^2. \tag{72}$$

## A.12 MULTILAYER COUPLED DYNAMICS

At the layer level, survival is not independent. Let $\Psi^{(k)}(t) = [\psi_i^{(k)}(t)]_{i \in \mathcal{N}_k}$ be the joint state of all neurons in layer $k$. We define the *inter-layer coupling operator*: We restrict attention to sensitivities between *activations* of adjacent layers. Let

$$J_{k \to k+1}(t) := \frac{\partial h^{(k+1)}(t)}{\partial h^{(k)}(t)} \tag{73}$$

denote the Jacobian mapping pre-activations/activations in layer $k$ to those in layer $k+1$ (evaluated pointwise and averaged over data when necessary) Li (2025); Laborieux & Zenke (2024). For neurons indexed $i \in \mathcal{N}_k, j \in \mathcal{N}_{k+1}$, we write the element-wise sensitivity as

$$\mathcal{C}_{k \to k+1}^{(i,j)}(t) := \frac{\partial a_j^{(k+1)}(t)}{\partial a_i^{(k)}(t)}. \tag{74}$$

To obtain a layer-level scalar measure that is robust to width, we define the *layer influence* by the width-normalized average operator norm:

$$\mathbf{M}_{k,k+1}(t) := \frac{1}{|\mathcal{N}_k||\mathcal{N}_{k+1}|} \sum_{i \in \mathcal{N}_k} \sum_{j \in \mathcal{N}_{k+1}} \left\| \mathcal{C}_{k \to k+1}^{(i,j)}(t) \right\|_{2 \to 2}, \tag{75}$$

where $\| \cdot \|_{2 \to 2}$ denotes the induced (spectral) norm of the scalar-to-scalar sensitivity (for scalar activations this is absolute value). Equivalently one may use the averaged Frobenius norm divided by $\sqrt{|\mathcal{N}_k||\mathcal{N}_{k+1}|}$ for implementation convenience Diakonikolas (2023); Laurent (2024); both variants are equivalent up to constant factors and we report which we use in experiments.

**Definition A.14** (Darwinian Flow Energy). The Darwinian flow energy is defined as

$$\mathcal{E}_{\text{Darwin}} := \sum_{k=1}^{D} \sum_{l=1}^{D} \int_{0}^{T} \mathbf{M}_{k,l}(t) \, \phi\big(\text{JS}(\rho^{(k)}(t) \, \| \, \rho^{(l)}(t))\big) \, dt, \tag{76}$$

or, alternatively,

$$\mathcal{E}_{\text{Darwin}}^{W} := \sum_{k,l} \int_{0}^{T} \mathbf{M}_{k,l}(t) \, \phi\big(W_1(\rho^{(k)}(t), \rho^{(l)}(t))\big) \, dt. \tag{77}$$

**Theorem A.15** (Coupled Survival Principle). *Suppose that for some $\mu > 0$ and a subset $\mathcal{S}^{(k)} \subseteq \{1, \ldots, n_k\}$ of survived neurons at layer $k$, the layer-to-layer coupling matrix $\mathbf{M}_{k,k+1}(t)$ satisfies*

$$\sum_{i \in \mathcal{S}^{(k)}} \mathbf{M}_{k,k+1}(i,j)(t) \geq \epsilon > 0, \tag{78}$$

*for all neurons $j$ in layer $k + 1$ and all sufficiently large $t$.*

*Then, there exists $\eta = \eta(\mu, \epsilon, \text{Lipschitz constants}) > 0$ such that at least an $\eta$ proportion of neurons in layer $k + 1$ achieve high fitness (survival).*

*Proof.* Positive lower bounds on coupling imply sustained energy inflow to downstream neurons. Via the Lipschitz continuity of the fitness function and the smoothness of the dynamics, survival of upstream neurons forces a positive measure of downstream neurons to cross the survival threshold. □

**Theorem A.16** (Global Convergent Specialization). *If the total Darwinian flow energy $\mathcal{E}_{\text{Darwin}} \geq \epsilon > 0$ is bounded away from zero and the fitness functions $\Phi_i^{(k)}$ are sufficiently smooth and Lipschitz continuous, then as $t \to \infty$, the proportion of neurons with fitness below any fixed threshold tends to zero.*

*Proof.* Construct a suitable Lyapunov function based on the sum over neurons of a decreasing convex function of their fitness values Chen (2024); Alfarano (2024). The positive lower bound on Darwinian flow energy ensures the Lyapunov function decreases over time, implying convergence to the set of neurons with high fitness. LaSalle's invariance principle excludes non-convergent oscillations. □

### A.13 Additional Experiments on Three-layer MLP-Net with MNIST

#### A.13.1 Dynamics Neuron Trajectory and Evolution Analysis.

Figure 4(a), top shows the PCA-projected trajectories of shallow-layer neurons across training. Survived neurons (green) follow relatively long and directed paths, indicating sustained representational change. Their motion exhibits fewer reversals than eliminated neurons (red), which instead display short and irregular trajectories, often collapsing toward the origin. This contrast is reflected quantitatively in Figure 4(c), top, where cumulative trajectory length grows steadily for survived neurons. The weight dynamics in Figure 4(d), top reinforce this pattern: survived neurons exhibit increasing $L_2$ norms of incoming weights, whereas eliminated neurons remain almost flat, suggesting a gradual withdrawal of representational capacity. Collectively, these results indicate that even in the shallow layer, gradient descent implicitly differentiates between neurons that maintain sustained alignment with the loss signal and those that do not.

In the middle layer (Figure 4(a), middle), the divergence becomes more pronounced. Survived neurons trace longer and more coherent trajectories, while eliminated neurons remain short and close to the origin. This is supported by Figure 4(c), middle, where the cumulative trajectory length of eliminated neurons grows at a substantially lower rate than that of survived neurons, already showing a marked slowdown by Epoch 2. Weight norms (Figure 4(d), middle) again show a separation,

with growth for survived neurons and almost stagnation for eliminated ones. Compared to the shallow layer, the selective bottleneck appears stronger: neurons that fail to establish early alignment with the optimization signal are rapidly marginalized. This suggests that middle-layer neurons, receiving both bottom-up and top-down gradients, undergo more stringent selection toward functional specialization.

The deep layer presents a smaller sample size, but a similar trend is observable. As shown in Figure 4(a), bottom, survived neurons follow more extended trajectories, while the eliminated neuron remains nearly static. Correspondingly, trajectory length (Figure 4(c), bottom) and weight norm evolution (Figure 4(d), bottom) both indicate continued adaptation for survived neurons but not for the eliminated one. Although the limited number of neurons precludes strong statistical claims, the observed divergence suggests that selection pressures persist even near the output. Importantly, this implies that architectural proximity to the loss signal alone does not guarantee survival; functional alignment remains necessary.

Overall, Figure 4 highlights a consistent layer-wise pattern: shallow-layer neurons exhibit the earliest divergence, middle-layer neurons experience intensified selection with clearer separation between survived and eliminated groups, and deep-layer neurons—though fewer—still reflect selective retention. These results support the view that neuron survival is not imposed externally but emerges from the training dynamics, with selection pressures varying in strength across depth.

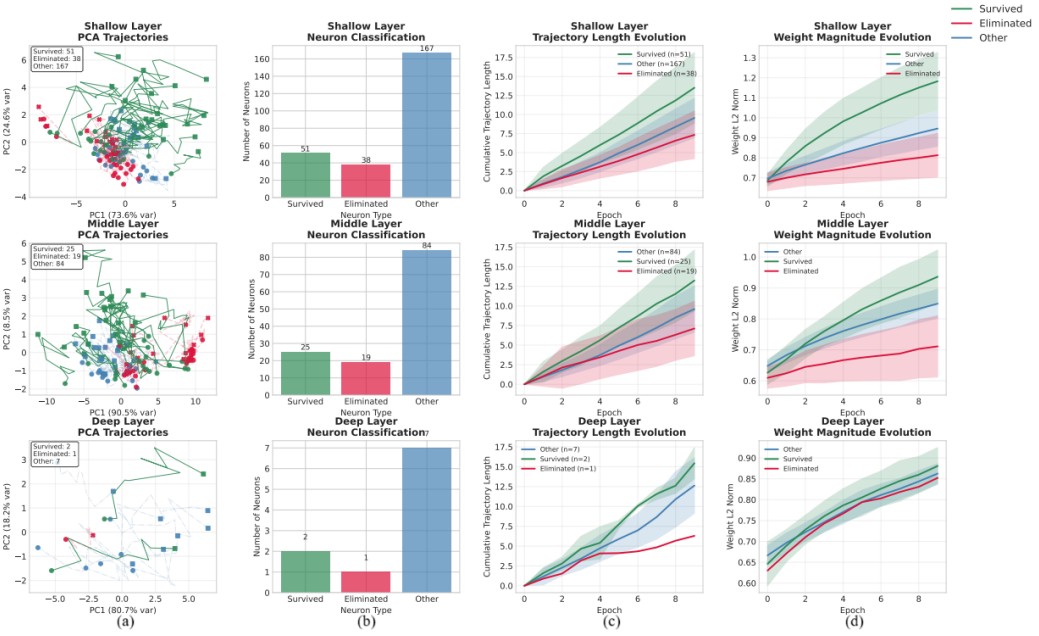

Figure 4: Dynamics Neuron Trajectory and Evolution Analysis on MNIST.

### A.13.2 STATIC PCA AND ACTIVATION EVOLUTION

Figure 5 (top-left) presents the final-epoch PCA projection of first-layer neuron activations. Neurons categorized as survived occupy relatively dispersed regions, often farther from the origin, which correlates with higher activation magnitude and greater variance. Eliminated neurons cluster near the origin, suggesting low-output states with reduced contribution to the representational space. The majority of neurons fall into the other category, exhibiting intermediate positions without clear clustering, reflecting heterogeneous or drifting roles during training. The activation-norm trajectories (Figure 5, bottom-left) provide a temporal view of this differentiation. Survived neurons increase their average norm across epochs, indicating sustained engagement with learning signals. Eliminated neurons, in contrast, display a gradual decline toward low, stable norms, consistent with functional silencing. The "other" group remains in an intermediate range, suggesting partial adaptation without clear reinforcement or suppression.

In the middle layer (Figure 5, top-middle), the PCA projection reveals that eliminated neurons are shifted toward the positive-PC1 periphery, while survived neurons occupy a broader and more heterogeneous region spanning both central and peripheral zones. The activation trajectories (bottom-middle) sharpen this divergence: survived neurons exhibit a sustained rise in activation norm, whereas eliminated neurons remain suppressed with only marginal growth. Taken as a whole, these patterns suggest that selection-like dynamics manifest most clearly in intermediate layers, where neurons are actively sorted into amplifying versus stagnant trajectories.

For the deep layer (Figure 5, top-right), the neuron count is small (only 2 survived and 1 eliminated), limiting statistical strength. The survived units exhibit higher final activation norms (bottom-right), whereas the eliminated unit declines toward a baseline. While this pattern resembles earlier layers, the small sample size precludes strong generalization.

Overall, the combination of static PCA projections and dynamic activation curves provides complementary evidence of neuron-level differentiation across depth. These results are consistent with the hypothesis that overparameterized networks allocate representational capacity unevenly, with some neurons reinforced while others become marginalized. However, the analyses are correlational and limited by dimensionality reduction and sample imbalance, particularly in deeper layers.

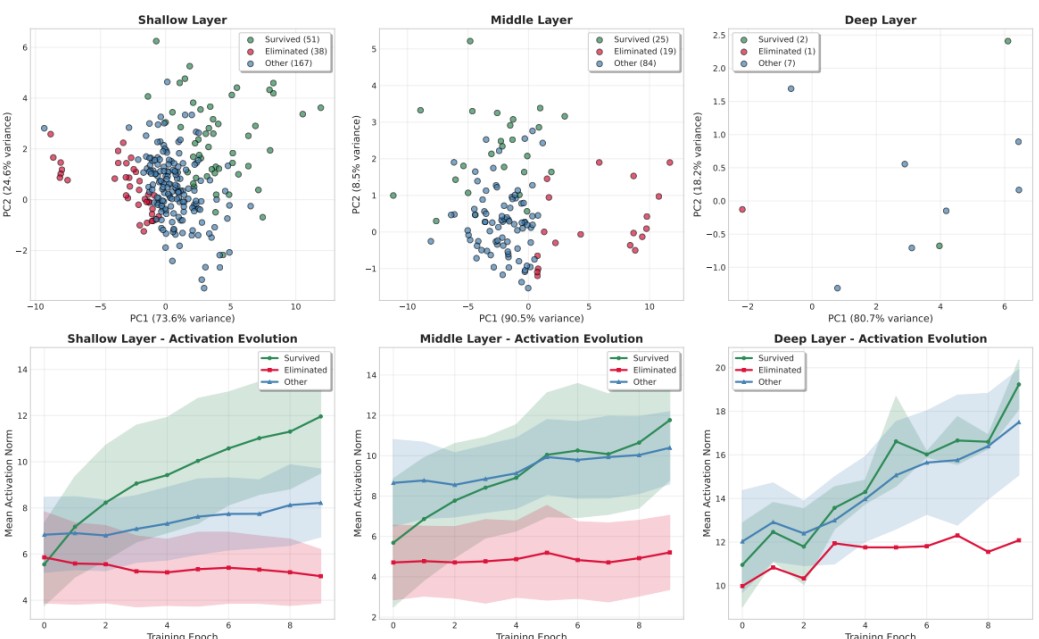

Figure 5: Static PCA and Activation Evolution on MNIST.

## A.14 ADDITIONAL EXPERIMENTS ON RESNET-18 WITH CIFAR-10

### A.14.1 DYNAMICS NEURON TRAJECTORY AND EVOLUTION ANALYSIS

The shallow layer dynamic PCA trajectories (Figure 6(a), top) show that neuron activations in early convolutional layers—often assumed to encode low-level, generic features—already exhibit signs of representational divergence. Survived neurons tend to follow more stable and moderately directed paths in the PCA manifold, with reduced dispersion over training, suggesting a gradual consolidation toward more compact representational regions. In contrast, eliminated neurons display more irregular trajectories, with frequent directional changes and less coherence, indicating comparatively unstable representational roles. This difference is also reflected in the cumulative trajectory length evolution (Figure 6(c), top): survived neurons maintain consistently higher cumulative movement compared to eliminated neurons, suggesting greater adaptability and sustained representational change across epochs. While the absolute gap is modest, survived neurons display more continuous directional displacement, whereas eliminated neurons tend to plateau earlier, consistent with a potential stagnation of their representational contribution. From a structural perspective, the

weight magnitude evolution (Figure 6(d), top) indicates that the convolutional filters corresponding to survived neurons generally retain slightly higher L2 norms throughout training, while those of eliminated neurons remain lower. This trend is consistent with the interpretation that neurons contributing more strongly to gradient pathways receive relatively greater synaptic reinforcement, whereas others undergo gradual attenuation. Collectively, these results suggest that even shallow layers are subject to competitive dynamics, where only subsets of neurons demonstrating sustained utility remain functionally active.

The middle layers serve as a transitional zone between low-level and high-level representations, and this role is reflected in the diversity of neuron trajectory dynamics. As shown in the dynamic PCA projections (Figure 6(a), middle), neurons in these layers exhibit heterogeneous representational paths over training. Survived neurons tend to follow longer and more coherent trajectories, often traversing distinct regions of the PCA manifold, suggesting a gradual alignment with intermediate-level features. By contrast, many eliminated neurons show less coherent movement, with shorter and more irregular trajectories, though some maintain moderate displacement comparable to the other group. The cumulative trajectory length curves (Figure 6(c), middle) provide quantitative support for these observations: on average, survived neurons reach greater cumulative lengths than eliminated or other neurons, reflecting more sustained representational plasticity. Eliminated neurons continue to grow but at a slower rate, with later signs of stagnation. A similar pattern is visible in the weight magnitude evolution (Figure 6(d), middle), where survived neurons exhibit slightly higher L2 norms than eliminated neurons. Although the difference is modest, its persistence across epochs indicates that neurons contributing more to the task tend to retain larger weight magnitudes. As a whole, these results suggest that the middle layers serve as a representational bottleneck where neurons undergo implicit selection, retaining those with flexible and task-relevant transformations.

In the deep layer, the contrast between neuron groups becomes more pronounced. As illustrated by the dynamic PCA trajectories (Figure 6(a), bottom), survived neurons follow long, smooth, and more aligned paths through representation space, frequently converging to structured low-dimensional subspaces. These neurons appear to encode abstract, class-discriminative information that supports final classification. In contrast, eliminated neurons reveal short, noisy, and non-convergent trajectories, often stagnating or oscillating without clear direction, suggesting limited long-term utility. This distinction is also evident in the trajectory length evolution (Figure 6(c), bottom), where survived neurons maintain the highest cumulative distances relative to eliminated neurons. These lengths reflect sustained representational change that tracks increasing class separability. Moreover, the variance among survived neurons is smaller, suggesting more constrained roles in the deep layer. The weight magnitude evolution (Figure 6(d), bottom) further highlights this separation: survived neurons retain high L2 norms, while eliminated neurons undergo progressive attenuation. The resulting divergence is strongest in this layer, consistent with stronger selective pressure as representations become more task-specific.

Overall, these findings are consistent with the framework of Neural Darwinism: across layers, neurons exhibit competitive dynamics shaped by their sustained utility. While shallow layers already show signs of divergence, the middle layers intensify selective processes, and the deep layers consolidate highly specialized neurons. The evidence from trajectory dynamics and weight evolution collectively supports the interpretation that representational selection operates hierarchically, shaping survival and elimination throughout the network.

### A.14.2 STATIC PCA AND ACTIVATION EVOLUTION

In Figure 7 left and bottom-left, the PCA projection (97.8% variance explained by PC1) shows that survived neurons occupy a relatively more compact region of the activation space, while eliminated neurons are scattered toward peripheral, low-density zones. Other neurons form a diffuse cloud spanning both regions. The activation evolution curves corroborate this structure: survived neurons sustain moderately higher activation norms with gradual stabilization, whereas eliminated neurons display persistently weak activations, and others remain intermediate. These patterns suggest that even at early layers—traditionally considered low-level feature extractors—there is already a degree of representational competition, consistent with the Neural Darwinism view that selection pressure operates from the outset of learning.

In Figure 7 middle and bottom-middle, the PCA embedding (94.2% variance explained by PC1) reveals a clearer differentiation than in shallow layers. Survived neurons cluster more tightly along

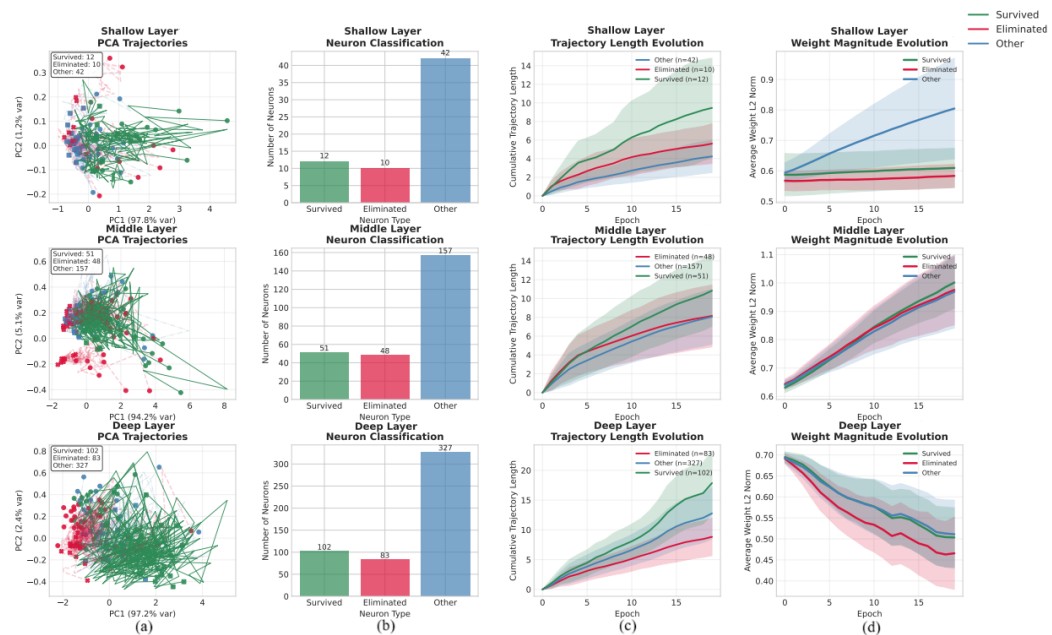

Figure 6: Dynamics Neuron Trajectory and Evolution Analysis on CIFAR-10.

dominant axes, while eliminated neurons are dispersed across orthogonal or low-density subspaces. Other neurons span an intermediate gradient, partially overlapping both groups. The activation dynamics mirror this structure: survived neurons maintain higher, stable activations, eliminated neurons steadily decline. These findings are consistent with the hypothesis that middle layers face stronger selective pressure, as they form an intermediate representational bottleneck where neurons must converge toward task-relevant manifolds to persist.

In Figure 7 right and bottom-right, in the final layer (97.2% variance explained by PC1), survived neurons are broadly distributed along the dominant axis but relatively compact along PC2, indicating alignment to a high-variance representational subspace. Eliminated neurons are concentrated in the lower-PC1 region, while others populate an intermediate zone overlapping both groups. The activation evolution curves reinforce this separation: survived neurons sustain the highest activation norms with relative stability, eliminated neurons remain consistently suppressed, and others occupy intermediate levels. Therefore, the static and dynamic views suggest that deep layers culminate the Darwinian competition, consolidating a high-utility representational manifold surrounded by marginal units.

### A.15 Additional Experiments on VGG-16 on CIFAR-100

#### A.15.1 Dynamics Neuron Trajectory and Evolution Analysis

In the shallow layer of Figure 8, the dynamic PCA trajectory analysis reveals early indications of neuronal differentiation consistent with the principles of Neural Darwinism. Survived neurons—characterized by relatively higher activation levels and modestly higher weight magnitudes—tend to originate near the PCA origin at the start of training and progressively diverge along more extended and directionally consistent paths in activation space (Figure 8(a), top). Their trajectories exhibit sustained cumulative displacement over the training epochs (Figure 8(c), top), suggesting continued adaptation. Although the paths are often noisy and irregular, the outward spread indicates a gradual specialization process that may enable distinct low-level feature subspaces to emerge under task-driven gradient signals. By contrast, eliminated neurons generally follow more compact trajectories, remaining closer to the origin and displaying shorter cumulative displacements (Figure 8(a,c), top). Their temporal variance is lower and their trajectory curvature less pronounced, implying reduced representational change. The L2 weight norms of this group are on average slightly lower than those of survived neurons, but the distributions remain strongly overlapping (Figure 8(d),

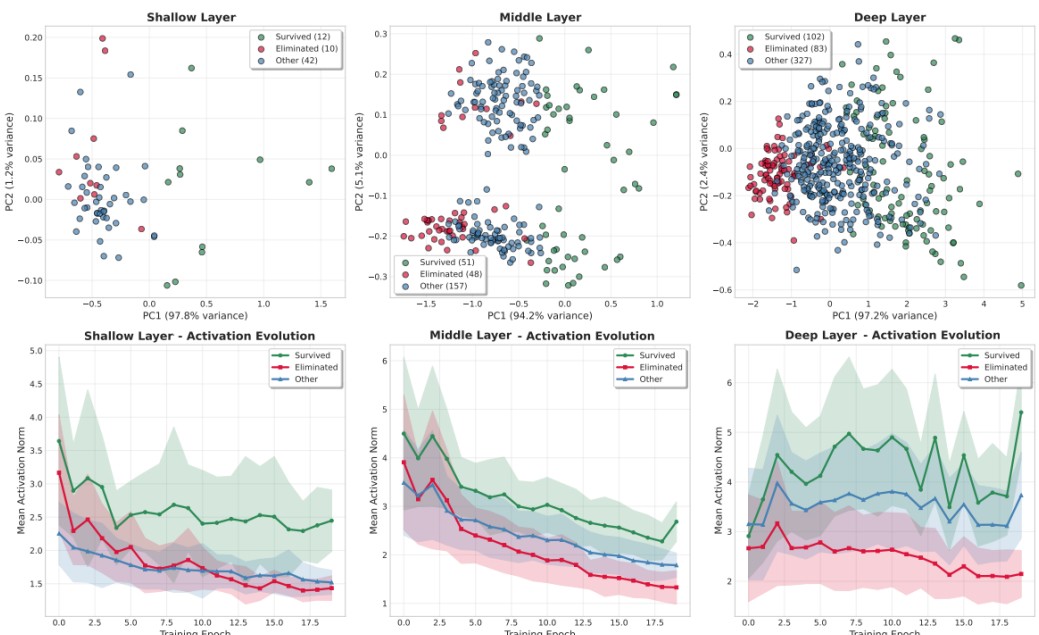

Figure 7: Static PCA and Activation Evolution on CIFAR-10.

top). While gradient flow is not directly quantified, the limited representational mobility is consistent with the interpretation that these neurons receive weaker or less task-relevant updates during training. The neurons classified as other occupy an intermediate position. Their trajectories are more diffuse and less directionally stable (Figure 8(a), top), with cumulative lengths that are broadly comparable to those of survived neurons but accompanied by larger variance (Figure 8(c), top). Some display periods of outward displacement before stabilizing, while others remain closer to the origin throughout. This heterogeneity suggests that they represent a transitional population whose role is not firmly consolidated within the finite training horizon. Overall, these patterns support a local form of Neural Darwinism: within the shallow layer, a subset of neurons progressively differentiates and maintains higher representational activity, whereas others remain less engaged and gradually lose relative influence. The emergence of such divergence close to the raw input highlights that selection pressures may act from the earliest stages of learning.

In the middle layer—where hierarchical abstractions become more pronounced—the selective dynamics appear intensified relative to the shallow layer. PCA trajectories (Figure 8(a), middle) show that many survived neurons diverge from the origin early and continue outward with sustained displacement, though their paths remain noisy and variable. While most neurons cluster near the PCA origin, a modest subset of survived neurons extends into more distinct regions of the projection space, suggesting partial occupation of differentiated representational subspaces. Eliminated neurons, by contrast, display shorter or less stable trajectories: some show brief excursions before returning toward the origin, whereas others remain in intermediate positions without consistent outward drift. The other neurons again form a heterogeneous group, with some traveling considerable distances but frequently changing direction, and others staying confined near the origin. Quantitatively (Figure 8(c), middle), survived neurons accumulate the greatest trajectory lengths by the final epoch, though the margin over other groups is modest (approximately 0.3–0.4 units). In terms of weight evolution (Figure 8(d), middle), all neuron types exhibit monotonic L2 norm decay, with survived neurons showing a slightly slower decline and thus ending with marginally higher magnitudes. This suggests that survival is associated with maintaining relatively stronger synaptic weights, though the effect size is small. Collectively, the middle layer illustrates an intensification of competitive dynamics, where survived neurons maintain more persistent representational mobility, eliminated neurons adapt weakly or transiently, and the majority of units remain in flux without converging to stable roles.

In the deep layer—the final fully connected stage before classification—the rate of representational change appears increased, consistent with a late-phase consolidation process. Survived neurons continue to accumulate trajectory length (Figure 8(c), bottom), but at a quicker rate compared to earlier layers. In the PCA projection (Figure 8(a), bottom), these neurons drift outward from the origin and follow moderately directed paths, with curvature and displacement gradually increasing over time. This pattern indicates partial stabilization, consistent with their role in encoding higher-level, semantically richer features that require fewer adjustments once tuned. Weight magnitude curves (Figure 8(d), bottom) similarly show that survived neurons maintain slightly higher norms than eliminated and other neurons, though the separation remains limited. Eliminated neurons in the deep layer exhibit shorter cumulative trajectory lengths and modestly lower weight norms. While some early movement is evident, their displacement growth slows considerably, and their PCA positions remain relatively central, indicating constrained representational change. The other group again occupies an intermediate position, with moderate representational shifts and weight growth, suggesting residual but limited contribution to the final predictive function.

In summary, these observations align with a Neural Darwinism perspective in which neuronal survival reflects continued representational mobility and modestly stronger synaptic weights, while elimination corresponds to reduced or transient adaptation. Importantly, the presence of a large heterogeneous other group underscores that selection pressure operates continuously, and many neurons remain in transition rather than converging to stable roles. The progression from shallow to middle to deep layers reflects a gradual sharpening of selection, culminating in a smaller set of stabilized neurons in the deepest layer.

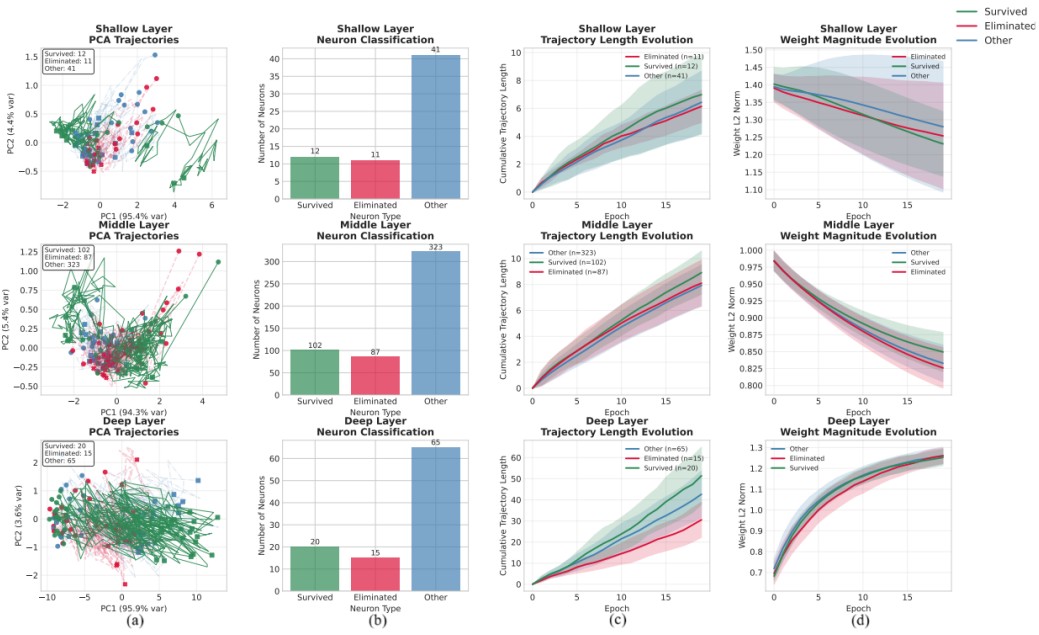

Figure 8: Dynamics Neuron Trajectory and Evolution Analysis on CIFAR-100.

### A.15.2 STATIC PCA AND ACTIVATION EVOLUTION

In the shallow layer, the final-epoch PCA projection in Figure 9 left shows that the first two principal components account for approximately 99% of the total variance (PC1: 95.4%, PC2: 4.4%), indicating that most inter-neuron activation variability can be represented in a low-dimensional subspace. Despite the limited receptive fields of early convolutional layers, survived neurons (green) occupy more peripheral regions of the PCA plane, with greater dispersion from the origin and from one another, suggesting a tendency toward differentiated feature sensitivities. By contrast, eliminated neurons (red) remain densely concentrated near the origin, reflecting low variance and limited representational differentiation. The activation evolution curves in Figure 9 bottom-left reinforce this observation: neurons with persistently higher activation norms tend to survive, while those with steadily declining norms move toward elimination. The distribution of survived neurons suggests

diversity in low-level tuning—potentially edges or localized textures—that broadens the expressive basis available for subsequent layers. While the pattern is not definitive, it is qualitatively consistent with a threshold-like competitive process, in line with selection mechanisms hypothesized in Neural Darwinism.

In the middle layer, the PCA projection in Figure 9 middle explains roughly 99% of the variance (PC1: 94.3%, PC2: 5.4%). Here, survived neurons (green) are broadly distributed across the PCA space, often forming multiple partially separated groups, whereas eliminated neurons (red) cluster tightly near the origin. The other group (blue) occupies an intermediate band, positioned between the high-variance survived regions and the low-variance eliminated cluster. Activation evolution patterns (Figure 9 bottom-middle) reveal that survived neurons maintain high and relatively stable activation norms, eliminated neurons exhibit a consistent decline, and others remain at intermediate levels with mild fluctuations. The spread of survived neurons across the PCA space suggests an increasing degree of representational diversification at this stage, corresponding to the formation of mid-level abstractions. The non-random structure—characterized by local coherence within groups and broader separation between groups—indicates systematic partitioning of representational space. The central concentration of eliminated neurons, coupled with their declining activations, is consistent with redundancy or reduced gradient flow, whereas the transitional behavior of the other group may reflect delayed specialization.

In the deep layer, corresponding to the final fully connected stage, the PCA projection in Figure 9 right shows that the first two principal components explain about 99% of the variance (PC1: 95.9%, PC2: 3.6%). This high concentration of variance suggests a compressed and highly structured representational space, consistent with the role of this layer in integrating features for classification. Survived neurons are predominantly located in peripheral regions of the PCA plane, often grouped into small clusters. The activation trajectories in Figure 9 bottom-right show that survived neurons maintain higher and often increasing activation norms across training epochs, indicating sustained engagement in the final decision space. By contrast, eliminated neurons cluster near the PCA origin and exhibit consistently lower activation magnitudes and slower growth, suggestive of early functional deactivation. Other neurons occupy intermediate positions, with activation dynamics reflecting transient or weak selectivity that does not consolidate into either survival or elimination.

Overall, the three-layer comparison in Figure 9 highlights a consistent pattern: variance in activations is concentrated in a few dominant dimensions, survived neurons occupy more dispersed regions and sustain higher activity levels, while eliminated neurons remain near the origin with declining activations. The other group exhibits transitional characteristics, reflecting instability or incomplete specialization. The combined static and dynamic views are qualitatively consistent with a selection-based process in which functionally distinctive neurons persist and redundant ones fade, echoing principles of Neural Darwinism.

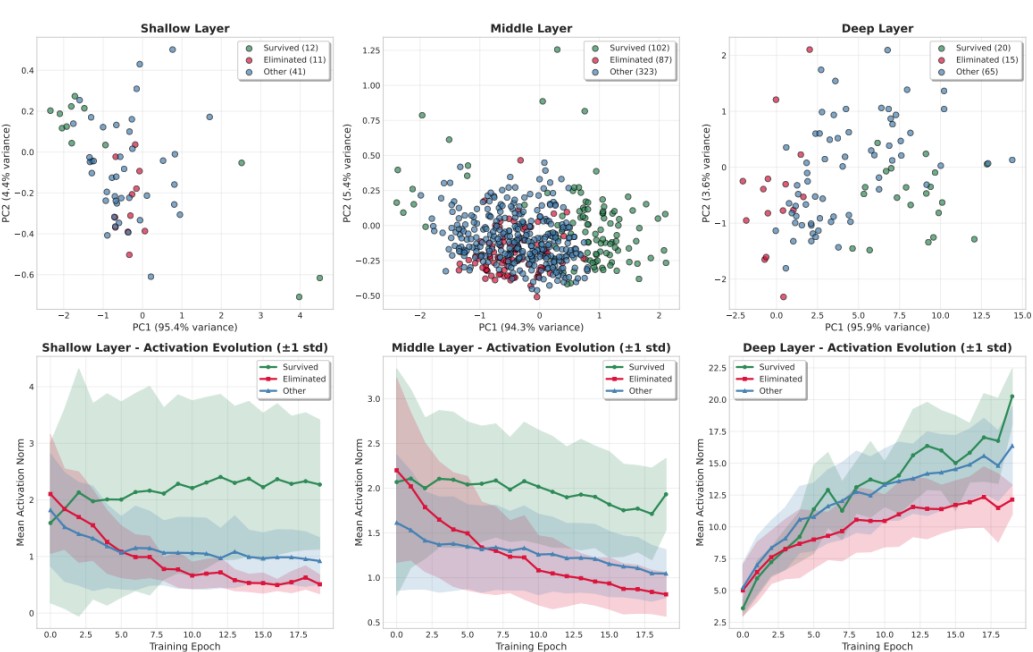

Figure 9: Static PCA and Activation Evolution on CIFAR-100.

