# OpenReview forum: "Trajectory-Based Neural Darwinism in Convolutional Neural Networks: Variation, Competition, and Selective Retention"
_ICLR.cc/2026/Conference — ICLR 2026 Conference Withdrawn Submission_

### Official Review · Reviewer_whuB · 2025-10-22

**Soundness:** 2
**Presentation:** 2
**Contribution:** 2
**Rating:** 2
**Confidence:** 3

**Summary:**

The paper introduces the Neuron Darwinian Dynamics System (NDDS), a framework for studying neuronal trajectories, based on the idea that neuronal populations evolve through competition and selection. The NDDS comprises of a neuron state vector which includes information about parameters, average activity, average gradient, and the entropy of the activity. This state vector is used to study representational and neuronal dynamics during training. In particular, the trajectory of each neuron in state space can be used to measure trajectory length, integrated entropy, and time-averaged ablation-based utility. These metrics are used to quantify evolutionary fitness. By comparing the fitness of each neuron to population statistics, neurons are classified as ‘survived’, ‘eliminated,’ or ‘other.’ The authors conduct an ablation study in CNNs trained on MNIST where they progressively remove neurons randomly and observe how accuracy and t-SNE projections change, finding that a large number (60%) can be removed with minimal change in performance. Next, using ResNet-50 trained on Tiny-ImageNet, they study the PCA, trajectory length, weight magnitude, and activation norm of neurons, comparing between the three classifications, finding that survived neurons appear to have greater representational changes in deep layers.

**Strengths:**

The initial motivation seems reasonable and interesting, given the literature reviewed. Indeed, demonstrating competitive ‘Darwinian’ dynamics could be a valuable contribution to the understanding of learning dynamics and representational learning. The framework introduced appears novel and is well-detailed in-text. They reproduce their experimental results across several architectures and datasets. The paper is well-written.

**Weaknesses:**

The paper claims to show Neural Darwinism in CNNs, where “neuronal populations evolve through variation, competition, and selective retention, forming stable yet adaptable circuits.” While they do provide some evidence that CNNs may rely on a subset of neurons, there’s no evidence provided to support that this is driven by neuron-level competition (or attempt to describe how this might occur), or that this forms stable and adaptable circuits. I would have liked to have seen mechanistic/theoretic analysis to support the authors’ claims.

The framework introduces many different metrics into its measure of neuronal trajectory and fitness, but it’s not clear whether they each contribute something meaningful to the network analysis. Furthermore, by concatenating all of these metrics into one vector, I’m concerned whether this might obfuscate understanding/the behavior of the network more than by studying each individually. I would suggest that these are more strongly motivated and that this aggregation is validated experimentally (i.e., compare the neural state vector with studying each metric separately). I’m also skeptical about the emphasis on a neuron-level analysis rather than at the population-level. There is increasing evidence that many ANNs use distributed neural representations, with perhaps some neurons specializing to dominant directions of the data (e.g., PCA) but others using a more distributed code (Elhage et al., 2022). Comparing “survived” versus “eliminated” neurons could potentially make it harder to capture this underlying behavior.

It's stated in-text that the survival criteria defined “creates an evolutionary-like selection pressure, where only the most informative and stable neurons persist.” But to clarify—this is a post-hoc classification? There’s also no discussion about the distinction between “eliminated” and “other” neurons in text. It’s not obvious that these categorizations are very informative for understanding the network, and what “other” neurons are supposed to be functionally.

The ablation experiment claims to “provide direct evidence for a Darwinian view of neural representations.” In my opinion, the experiment only provides evidence that there are redundant or inessential neurons in the CNN that can be removed. This does not necessarily imply, for example, competition and selection during training-- the network may be overparameterized and have information repeated in several neurons. Furthermore, the experiment doesn’t use the framework that the authors have just constructed, but makes claims about the “Darwinian” perspective. It claims that “once the ablation encroaches upon the Darwinianly selected subset of neurons… both accuracy and representation quality collapse.” In the main text, it states that these neurons are randomly ablated, which would suggest that this subset of neurons is not some “selected, specialized” subset of neurons, but a random assortment. To make a stronger claim about their framework, the authors could ablate the “survived” neurons (according to their classification) to show that they will cause a drop in performance, as well as the “eliminated” neurons and show that performance is retained. Furthermore, the effect of removing neurons without a substantial change in performance has been reported before (for example, Casper, Boix, et al., 2021; there may be others as well).

As I said with the previous experiment, it would be useful to provide an ablation experiment to support the claim that “eliminated” neurons are actually functionally eliminated in the network (and perhaps “other” neurons, depending on what their function is). If comparing between the different categorizations of neurons in terms of their metrics, there should be measures of statistical significance reported; visually, it’s not clear whether “trajectory length” or “weight magnitude” is significantly different between the different classes. It’s not clear how the cross-layer analyses are benefitted by the NDDS compared to performing PCA or other analyses on these metrics alone. In particular, it’s not clear to me what is explained by this experiment that hasn’t already been described by previous work that has looked at manifold dimensionality in different layers of CNNs (for example: Recanatesi, Farrell, et al., 2019; Ansuini et al., 2019; Cohen et al., 2020; Sorscher et al., 2022). Overall, in this experiment, it’s not clear what novel contribution the NDDS provides to the understanding of CNNs.

In general, I think the claims of this paper could be softened more. There could be more theoretical evidence/toy model experiments that something like competition is occurring, or more carefully designed experiments that show this and show that the framework faithfully captures this behavior in a way that is more informative/useful than the chosen metrics used separately. The choices made in creating the framework could also be motivated and discussed more.

**Questions:**

Questions (other than the points made above):

1.	What does the “stochasticity” metric refer to? Furthermore, in Lemma 3.11, is it reasonable to assume that stochasticity grows during training? I would presume the opposite.

2.	Can you discuss your choice of definition for fitness? I understand utility, but can you motivate/discuss stochasticity and entropy more?

3.	How are the fitness parameters chosen? It’s not clear to me from the text and also not clear to me how different selections might impact the results differently. Could you discuss this more?

4.	What does stability refer to in the text?

5.	How are “other” and “eliminated” neurons defined? How are they functionally distinguished in your framework? And why are there more “other” neurons than any other?

Additional suggestions not already mentioned above: include a contribution statement; more detailed figure captions; specify more experimental details—like what shallow, middle, and deep layers specifically refer to in a ResNet-50; define stochasticity metric (it’s not defined in-text); explain the reasoning behind choices made in creation of the framework; the y-axis of the neuron classification plot says “number of features”, but I assume you meant number of neurons. Also, Saxe, Sodhani, and Lewallen, 2022 might be useful for thinking about the learning dynamics and potentially competition.

---

### Official Review · Reviewer_Ddao · 2025-10-29

**Soundness:** 1
**Presentation:** 2
**Contribution:** 2
**Rating:** 2
**Confidence:** 2

**Summary:**

Focusing on CNNs, this paper draws parallels between Neural Darwinism and the optimisation process of neural networks. More precisely, they identify that the different neurons follow dynamics of survival and selection, with some neurons being discarded over time while others retain most of the discriminative information.

The authors first propose a theoretical framework linking neural network training with Neural Darwinism. To do so, they develop a “fitness” metric which balances neuron utility, entropy and stochasticity. Neurons with high fitness are categorised as “survived”. They finalise their theoretical investigation by showing a link between their fitness metric and gradient-variance contribution.

They follow-up with a few empirical evaluations of their theory. In the first experiment, they show that low proportions of neuron ablations do not impact representation quality much, whereas  higher proportions can more importantly hurt them. The second experiment studies the evolution of survived vs. eliminated neurons, looking at the PCA of their representations, their trajectory lengths and their magnitude. The final experiment studies the same neurons, this time using static PCA at the last epoch and the evolution of their activations.

**Strengths:**

The analogy between Neural Darwinism and neural network optimisation is compelling and theoretically promising. This theory could be a good explanation for several key properties of neural networks: the lottery ticket hypothesis, neuron pruning etc.

**Weaknesses:**

In its current form, I unfortunately think that this paper has multiple flaws which hurt its clarity and soundness.

W1: There are, I believe, some elements missing in the different definitions in Section 3 which make it difficult to understand the paper. The most important of those is that the definition of \$\mathcal{S}_i^{(k)}\$ looks to be missing. Because of this, evaluating the core metric, evolutionary fitness, is difficult.

W2: I have difficulties linking the theoretical results in Theorem 3.13 with the main conclusion of the paper, i.e. that “neurons are no longer seen as static units with fixed importance, but as evolving entities competing for survival through their trajectory length, stability, and entropy.” From my understanding, the authors develop the fitness metric as a time-averaged measure that increases with neuron utility and entropy, and decreases with stochasticity. They then prove that based on some assumptions, there exists some threshold such that above it, fitness implies a minimum level of gradient-variance contribution. While Theorem 3.13 establishes a relationship between fitness and gradient-variance contribution, the leap to framing neurons as “evolving entities competing for survival” is not sufficiently justified. A clearer connection between the mathematical results and the biological analogy would strengthen the paper’s argument.

W3: The empirical evaluations, while interesting, require more clarification and links to the theory to fully support the paper's claims.
* In Section 4.1, it is unclear which neurons are being removed, is it based on the fitness metric? On what is the t-SNE computed? These kinds of pruned neural networks are known in the literature, how do they relate to Neural Darwinism?
* In Section 4.2, while it is clear what neurons are labelled as “survived” from Equation 17, I have not seen the other 2 categories defined; this is problematic since the largest category are the “Other” neurons.
* I don’t necessarily agree with the qualitative arguments of the authors such as “survived neurons generally trace longer and more directionally consistent paths”, as from Figure 2a it also looks like red trajectories are both long and parallel to one another.
* Regarding Figures 2c and 2d, the width of the confidence intervals makes it impossible to make any definitive conclusions.
* Finally, in Figure 3, I get that the neurons with higher fitness also have higher activation values; this would be a good justification for using activation-based pruning, which is well established by itself.

**Questions:**

It is generally unclear to me how the theory developed by the authors and the experiments they run are related to their main claim. I understand how they can use the fitness metric to single out the most impactful and information-rich neurons and put them into the “survived” category. I don’t understand, however, how it relates to “evolving entities competing for survival”.

---

### Official Review · Reviewer_DKY9 · 2025-11-01

**Soundness:** 1
**Presentation:** 1
**Contribution:** 1
**Rating:** 0
**Confidence:** 4

**Summary:**

The paper adapts Edelman’s theory of Neural Darwinism to interpret the dynamical trajectories of neurons in an artificial neural network during training. First, a criterion for evolutionary fitness is defined, which takes a neuron's trajectory (weighted across parameter space, activation space, and some other things), and scores it by a weighted average over some metrics derived from the trajectory (these metrics generally describe the neuron's utility). The fitness criterion is proved to imply high gradient variance contribution, and this is related to a signal-to-noise criterion (theorem 3.13). Experiments show 1) the effects of pruning on the distribution of neuron representations, 2) differences in trajectories of neurons classified according to the fitness criterion, and 3) differences in final neuron states for the same classifications of neurons.

**Strengths:**

There is an interesting idea about classifying trajectories using evolutionary ideas from neuroscience. Also, the idea of looking at trajectories as dynamical systems is a rich and promising area of study.

**Weaknesses:**

At a high level, I am not sure how biological evolution (which has selective pressures such as death) maps onto neural network training.

The theory is not well explained - the motivation and intuition is unclear both overall for fitness $\phi$, and for the specific definitions and steps given in section 3 (for example, what is trajectory length used for? What is the point of lemma 3.11? Why do we care that $\phi$ goes to $-\infty$ in particular, and not just that it is smaller than some bound or than the average over all neurons?).

The fitness value $\phi$ is quite confusing - is this a theoretically derived quantity (i.e. falls out of the "natural" laws present), or an empirically motivated measure (i.e. designed by humans to have some practical utility)? Why does increasing entropy (positive contribution of $\mathcal{I}$) indicate greater fitness? The dependence of $\phi$ on what I presume are hyperparameters $\alpha, \beta, \gamma$ is not explained. Also, why are the activation values $\mu$ and the entropy $\mathcal{I}$ involved in the neuron trajectory $\psi$ (equation 2)? In SGD the evolution of the parameters only depends on the gradient, so these other terms seem superfluous (unless we are concerned with Adam, in which case the work should say so explicitly). Not to mention, the block-wise scaling matrix $D$ (equation 10), in addition to being yet another hyperparameter, is not well motivated.

The experiments in section 4 simply do not support section 3's derivations. For instance, I really don't understand how figure 1 has anything to do with the theory established in section 3. Section 4.2's PCA trajectories don't show any convincing trends.

Specifically for 4.2.1 and figure 2:
- PCA could be separating the trajectories simply by the way they are defined - e.g. if I say the top 50% of some linear combination of variables belongs to class A and the rest to class B, then class A and B will most likely be visibly split when taking PCA over the same variables, simply because I defined the classes that way.
- number of features in "other" dwarfs survived and eliminated
- no trends in trajectory length (they all fall within each other's confidence bounds!)
- no or confusing trends in weight magnitude (why would eliminated neurons have *higher* L2 norm than survived neurons? If eliminated neurons have no effect on the activations, and thereby no gradient, weight decay should make their weights smaller)

I will gloss over 4.2.2's issues but they are similar to those in 4.2.1.

Other issues:
- citations in the introduction need curation as many of them are only tangentially related to the present work.
- please cite Edelman again on line 38
- when referring to the appendix in the main text, links should be provided to specific sections
- the symbol in equation 11 (blackletter H?) could be replaced with something more ordinary and easy to read
- integrated entropy is not clearly defined (equation 11)
- S is not defined in equation 15 (also SD wasn't introduced explicitly)
- figure 2 is far too vague: what are "shallow", "middle", and "deep" layers; and what are "survived" versus "eliminated", or peculiarly, "other" neuron trajectories?

**Questions:**

In Lemma 3.11, are there limits to how small or large $\delta$ can be in equation 19? In principle I could imagine taking $\delta$ to be infinitesimally small in the hopes of catching the network in a stable moment (since training is noisy), thereby making the quantity in equation 19 as small as I wish.

Why is T-SNE used in figure 1 and PCA in figures 2 onwards?

---

### Official Review · Reviewer_wKCD · 2025-11-04

**Soundness:** 3
**Presentation:** 3
**Contribution:** 3
**Rating:** 6
**Confidence:** 4

**Summary:**

This paper explores whether selection like dynamics analogous to biological Darwinism emerge in artificial neural networks during training. Motivated by Edelman’s Neural Darwinism, the authors propose a theoretical and empirical framework, Neuron Darwinian Dynamics System (NDDS), to quantify neuron level variation, competition, and selective retention. NDDS models each neuron as an evolving entity characterized by activation trajectories, entropy, gradient statistics, and ablation-based utility. These quantities define a composite “fitness” score that identifies survived and eliminated neurons throughout training.

The authors apply NDDS across multiple architectures (MLP, ResNet-18, VGG-16, ResNet-50) and datasets (MNIST, CIFAR-10, CIFAR-100, Tiny-ImageNet). The framework includes dynamic analyses (trajectory length, activation entropy, weight evolution) and ablation studies to measure resilience under neuron removal. Results show that CNNs exhibit consistent Darwinian patterns: redundancy at early stages provides robustness, while deeper layers consolidate selective subsets of high-fitness neurons responsible for stable representations. Ablation experiments reveal sharp performance collapse beyond a critical neuron-removal threshold, confirming selective retention. The paper argues that this process parallels biological selection and provides a unifying lens to study representational specialization and robustness in deep networks.

**Strengths:**

- Conceptually novel framing: The paper’s integration of evolutionary theory with neural network dynamics provides a fresh and thought-provoking perspective on representational learning.

- Unified mathematical formalism: NDDS offers a systematic way to quantify neuron level dynamics through trajectory length, entropy, and ablation utility, bridging dynamical systems theory with information-theoretic measures.

- Comprehensive empirical validation: The study spans multiple architectures and datasets, demonstrating consistent neuron-level selection patterns and robustness trends.

- Clear empirical evidence of selective retention: The ablation experiments and trajectory visualizations vividly show redundancy, competition, and collapse once critical neurons are removed.

- High interpretability: The use of PCA trajectories, activation maps, and visual analyses across layers provides intuitive and interpretable evidence of selective neuronal dynamics.

**Weaknesses:**

- Limited theoretical depth: While NDDS is mathematically formalized, the theoretical contributions stop short of deriving predictive or generalizable results beyond descriptive dynamics.

- Ambiguity in biological analogy: The Darwinian framing, while appealing, may overextend the biological metaphor; the evidence supports competition and selection but not true evolutionary inheritance or adaptation.

- Lack of causal verification: The link between neuron “fitness” and actual causal contribution to network output remains correlational rather than experimentally validated.

- Overemphasis on CNNs: The exclusion of transformer architectures limits generality, especially given the field’s shift toward attention-based models.

- High conceptual complexity: The paper introduces numerous quantities (entropy, trajectory length, ablation utility) and theoretical assumptions, which, while detailed, may obscure the core contribution for a general audience.

**Questions:**

NA

---

### Note · Authors · 2025-11-12

I have read and agree with the venue's withdrawal policy on behalf of myself and my co-authors.